

# Topological defects in Floquet circuits

**Mao Tian Tan[1,2]⋆, Yifan Wang[3] and Aditi Mitra[2]**

**1** Asia Pacific Center for Theoretical Physics, Pohang, Gyeongbuk, 37673, Korea
**2** Center for Quantum Phenomena, Department of Physics, New York University,
726 Broadway, New York, NY, 10003, USA
**3** Center for Cosmology and Particle Physics, Department of Physics, New York University,
726 Broadway, New York, NY, 10003, USA

⋆ maotian.tan@apctp.org

## Abstract

We introduce a Floquet circuit describing the driven Ising chain with topological defects. The corresponding gates include a defect that flips spins as well as the duality defect that explicitly implements the Kramers-Wannier duality transformation. The Floquet unitary evolution operator commutes with such defects, but the duality defect is not unitary, as it projects out half the states. We give two applications of these defects. One is to analyze the return amplitudes in the presence of "space-like" defects stretching around the system. We verify explicitly that the return amplitudes are in agreement with the fusion rules of the defects. The second application is to study unitary evolution in the presence of "time-like" defects that implement anti-periodic and duality-twisted boundary conditions. We show that a single unpaired localized Majorana zero mode appears in the latter case. We explicitly construct this operator, which acts as a symmetry of this Floquet circuit. We also present analytic expressions for the entanglement entropy after a single time step for a system of a few sites, for all of the above defect configurations.

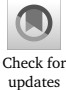

# 1  Introduction

Quantum circuits are an active area of research as they provide a tractable way to simulate dynamics of quantum systems with applications to experimental platforms such as trapped ions [1–3], Rydberg atoms [4], and Noisy Intermediate Scale Quantum (NISQ) devices [5–8]. Quantum circuits have been used to explore diverse phenomena such as thermalization [9,10], entanglement growth [11–17], quantum chaos [18–22], discrete time-crystals [23], information scrambling [24–27], and topological phases [28,29]. Many of these quantum circuits generalize Floquet (temporally periodic) circuits in various ways, most notably by introducing randomness in either the temporal or spatial direction, or both [30–32]. These generalizations of Floquet circuits have also been used to identify new forms of quantum criticality such as measurement-induced phase transitions [33–42] where the measurements introduce randomness in the temporal and spatial direction, with the projective measurements also making the quantum circuits non-unitary.

A well-known idea in equilibrium statistical mechanics is that an enhanced understanding of a model can be achieved after its dual has been identified and understood. For example, the study of dualities has led to an improvement in the understanding of phases of matter and the critical points that separate them [43]. This view, however, has not propagated into the literature on non-equilibrium systems. This paper aims to remedy this by making a first step in understanding the dual of a Floquet Ising chain. Mapping a model to its dual in equilibrium statistical mechanics occurs via, for example, the Kramers-Wannier transformation. One way of implementing this transformation is via a *topological duality defect*. Such a defect separates a (spatial or temporal) region from its dual, but the physics is independent of deformations of its location. In this paper we generalize these notions to Floquet systems.

In particular, we introduce an Ising Floquet circuit that hosts topological defects. Perfect spatial periodicity and time periodicity can be destroyed, but in a universal way so that the

physics is independent of deformations of the spatial or temporal locations of the defects. We present these Floquet circuits in the context of the transverse-field Ising chain, including inhomogeneous couplings and Floquet driving. We show that this system can host two types of topological defects, spin-flip and duality, just as in the equilibrium case [44]. These generalize those in the Ising conformal field theory (CFT) [45–47] in several ways: not only are they exactly topological on the lattice, but they exist for all couplings, not just critical ones. Our results here further generalize them to real-time Floquet evolution.

Each such topological defect can be applied to a Floquet circuit in two ways. Stretching the defect in the spatial direction allows us to define the defect-creation operator commuting with the Floquet unitary. These operators obey the same fusion rules as the Ising anyons [44, 48], and imply that the duality defect-creation operator is not unitary, but rather annihilates all states odd under spin-flip symmetry. The second application is to stretch the defect in the temporal direction, leading to generalized twisted boundary conditions. Such boundary conditions are special in that they preserve (a modified) translation invariance as the partition function is independent of the location of the twist/defect. Here, the two types of defects result in anti-periodic and duality-twisted boundary conditions [49–51].

So far, the Floquet literature has only studied open or periodic boundary conditions, and not twisted boundary conditions like the ones studied in this paper. In addition, while the effects of defects on the ground state entanglement entropy have been explored recently [52–54], these defects will not lead to interesting changes in the entanglement entropy in a periodically driven system in the absence of disorder since these systems will simply heat up, with the entanglement entropy evolving towards the thermal entropy that scales as the length of the sub-system.

One striking consequence of duality-twisted boundary condition is an isolated localized Majorana zero mode. This lone mode is unlike the pair of edge "strong zero modes" encountered in the transverse field Ising model with open boundary conditions, where the modes anti-commute with a symmetry of the Hamiltonian [55] or the Floquet unitary [56, 57]. In contrast, the Majorana zero mode in this paper is a symmetry of the system. We obtain the Majorana operator in an exact closed form by a direct computation, as opposed to the calculation of winding numbers as is commonly done in the study of Floquet Symmetry Protected Topological (SPT) phases [58, 59]. The twisted boundary conditions affects both the number of localized Majorana modes and their quasi-energies in comparison to defectless Floquet SPT models.

The paper is organized as follows. In Section 2 we introduce the two types of defects and describe the corresponding defect-creation operators. We summarize the commutation relations that establish their topological properties, following closely the analogous treatment presented in [44] for the classical Ising model and the static quantum spin chain. In Section 3 we numerically simulate the system in the presence of topological defects, highlighting the effect of the fusion rules on a return amplitude. In Section 4 we explain how topological defects allow the Floquet circuit to have anti-periodic and duality-twisted boundary conditions, and display amplitudes for the former. We show in Section 5 how a localized Majorana fermion emerges for duality-twisted boundary conditions. We give an explicit analytic construction and also present numerical studies of the auto-correlation function. Our conclusions and outlook are presented in Section 6. In Appendix A we demonstrate the effect of these defects on the entanglement entropy of a small system of four sites, where two sites host the physical qubits and the other two are the empty or dual sites. In Appendix B, we provide some intermediate technical steps. While the main text studies topological defects on the lattice, in Appendix C we describe their various aspects in the continuum. In particular, we review topological defects in the Ising CFT and their descriptions in terms of a single $1+1d$ Majorana fermion, and discuss the Majorana zero mode from the duality twist in the continuum theory.

## 2  Lattice topological defects

In this section, we review the relevant results for topological defects in equilibrium Ising systems [44], and generalize the notion to the Floquet Ising model. These defects break the homogeneity of the system, either temporally by acting on the system at a certain time, or spatially via changing the boundary condition at a certain point in space. However, they are topological because they can be freely deformed throughout the quantum circuit without changing amplitudes. The treatment of defects here is similar to that of equilibrium statistical-mechanical models; the main difference being an additional factor of $i$ coming from working in real time as opposed to Euclidean.

We study a system comprised of $L$ qubits/Ising spins, each with a local Hilbert space $\mathbb{C}^2$. For implementing the Kramers-Wannier duality, it turns out to be convenient to place the qubits on every other site of a chain of $2L$ sites labelled $j = 0, 1, 2, \ldots, 2L-1$. When the qubits are located on the "even" sublattice with $j = 2r$ for $r$ integer, the quantum states live in the Hilbert space $\mathcal{H}_{\text{even}}$, while when on the "odd" sublattice with $j = 2r-1$ they live in $\mathcal{H}_{\text{odd}}$. The basis elements for $\mathcal{H}_{\text{even}}$ are labelled by $|h_0 \sigma h_2 \sigma \ldots h_{2L-2}\sigma\rangle$ and for $\mathcal{H}_{\text{odd}}$ are $|\sigma h_1 \sigma h_3 \ldots \sigma h_{2L-1}\rangle$, where $h_j \in \{0, 1\}$ labels the qubit state, and sites without a qubit are labelled with a "state" $|\sigma\rangle$. The Pauli matrices acting non-trivially on the $j$th qubit are denoted by $X_j$, $Y_j$ and $Z_j$ and act as $Z_j|h_j\rangle = (-1)^{h_j}|h_j\rangle$, $X_j|h_j\rangle = |1-h_j\rangle$. We impose periodicity so that the site $2L + j$ is identified with the site $j$.

The basic building block of the periodically driven Ising model is a three-site gate obtained by acting on these qubits with the transverse-field and Ising-coupling unitaries

$$W_j^X(u_j) = e^{-iu_j X_j}, \qquad W_j^{ZZ}(u_j) = e^{-iu_j Z_{j-1} Z_{j+1}}, \tag{1}$$

where $W_j^X(u_j)$ is a transverse field unitary with a field strength $u_j$ that acts on the qubit at site $j$, while $W_j^{ZZ}(u_j)$ is an Ising coupling unitary that acts on the sites $j-1$ and $j+1$ with an Ising coupling given by $u_j$. The full Floquet unitary evolution operators are

$$U_{\mathbb{1},\text{even}} = \left(\prod_{j=0}^{L-1} W_{2j}^X(u_{2j})\right)\left(\prod_{j=0}^{L-1} W_{2j+1}^{ZZ}(u_{2j+1})\right), \tag{2a}$$

$$U_{\mathbb{1},\text{odd}} = \left(\prod_{j=0}^{L-1} W_{2j}^{ZZ}(u_{2j})\right)\left(\prod_{j=0}^{L-1} W_{2j+1}^X(u_{2j+1})\right), \tag{2b}$$

acting on $\mathcal{H}_{\text{even}}$ and $\mathcal{H}_{\text{odd}}$ respectively. The difference between placing the spins on the even or odd sites shows up in the order in which the operators containing the transverse magnetic field and the Ising coupling unitaries are applied.

A convenient pictorial presentation of the matrix elements of the building blocks (1) is given by

$$\langle \sigma h_j' \sigma | W_j^X(u_j) | \sigma h_j \sigma \rangle, \tag{3a}$$

$$\langle h_{j-1} \sigma h_{j+1} | W_j^{ZZ}(u_j) | h_{j-1} \sigma h_{j+1} \rangle, \tag{3b}$$

where the variables $h_{j-1}, h_j, h'_j, h_{j+1} \in \{0, 1\}$ and the unlabelled sites correspond to the label $\sigma$. For later convenience we include notation where the black dots indicate additional weights depending on the degree of freedom at the site [44, 48]

$$\bullet := d_i, \qquad \blacktriangle = \blacktriangledown = \sqrt{d_i}, \tag{4}$$

where

$$d_i := \begin{cases} d_\phi = 1 & \text{for } h_i = 0, 1, \\ d_\sigma = \sqrt{2} & \text{for } \sigma \text{ sites.} \end{cases} \tag{5a}$$
$$\tag{5b}$$

A picture depicting a single time step of our Floquet circuit is shown in figure 1. It represents the matrix element $\langle h'_0 \sigma h'_2 \sigma \dots h'_{2L-2} \sigma | U_{\mathbb{1},\text{even}} | h_0 \sigma h_2 \sigma \dots h_{2L-2} \sigma \rangle$. The analogous picture for $U_{\mathbb{1},\text{odd}}$ has the labels on the odd sites. The above setup can also be viewed as an interaction round-a-face model which are statistical mechanical models where the degrees of freedom live on the vertices of a square lattice and the interactions between them are defined by the plaquettes of the square lattice [60–65].

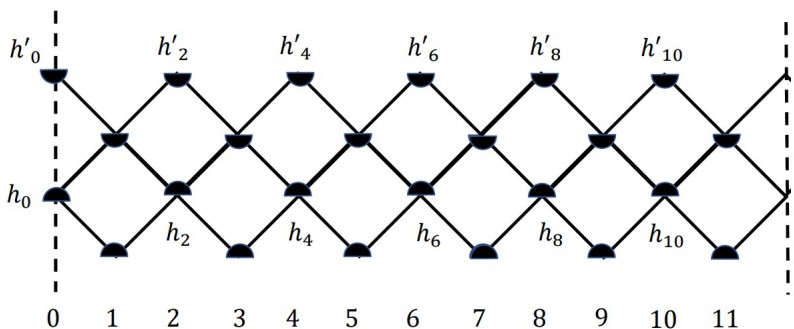

Figure 1: Diagram depicting the matrix elements of the Floquet Ising unitary $U_{\mathbb{1},\text{even}}$ for a single time step acting on $\mathcal{H}_{\text{even}}$ with $L = 6$. The indices $h_i$ ($h'_i$) label the states $0, 1$ of the incoming (outgoing) states of the Floquet unitary. The unlabelled vertices correspond to the label $\sigma$.

The simplest defect possible in the Floquet circuit implements the $\mathbb{Z}_2$ spin-flip symmetry. The corresponding operator $D_\psi$ creates a *spin-flip defect* across space, and acts on the odd and even Hilbert spaces as

$$D_{\psi,\text{odd}} = \prod_{r=0}^{L-1} X_{2r+1}, \qquad D_{\psi,\text{even}} = \prod_{r=0}^{L-1} X_{2r}. \tag{6}$$

It is easy to check that this defect commutes with the Floquet unitary: $D_\psi U_{\mathbb{1}} = U_{\mathbb{1}} D_\psi$. (When we omit the even and odd subscripts, it means that the relation holds for either case.) This "space-like" defect is comprised of spin-flip defect gates with matrix elements

$$\begin{array}{c} h'_j \\ \boxed{\phantom{xx}} \\ h_j \end{array} = \begin{array}{c} h'_j \\ \boxed{\phantom{xx}} \\ h_j \end{array} = \frac{1}{2^{1/4}} \delta_{h'_j, 1-h_j}. \tag{7}$$

Thus the matrix elements of $D_{\psi,\text{even}}$ are pictured by

$$\tag{8}$$

and likewise for $D_{\psi,\text{odd}}$.

The duality defects are much less obvious, and much more interesting. They implement Kramers-Wannier duality [61, 66], which in general maps a classical statistical-mechanical model on a graph to one on the dual graph with the same partition function. In the two-dimensional classical Ising model and the corresponding quantum chain, a convenient way of implementing the duality is via a topological defect [49–51]. The same goes for our Floquet setting. The duality defect gate is defined in terms of an operator $L_{j,j+1}$ with the matrix elements

$$\langle h'_j \sigma | L_{j,j+1} | \sigma h_{j+1} \rangle := \frac{1}{\sqrt{2}} (-1)^{h'_j h_{j+1}} , \tag{9a}$$

$$\langle \sigma h'_{j+1} | L_{j,j+1} | h_j \sigma \rangle := \frac{1}{\sqrt{2}} (-1)^{h_j h'_{j+1}} . \tag{9b}$$

The product of the duality defect gates produces a duality-defect creation operator $D_\sigma$ mapping between $\mathcal{H}_{\text{even}}$ and $\mathcal{H}_{\text{odd}}$. Its matrix elements are

$$= \left( \frac{1}{\sqrt{2}} \right)^L (-1)^{\sum_{r=0}^{L-1} h'_{2r+1}(h_{2r}+h_{2r+2})} , \tag{10a}$$

$$= \left( \frac{1}{\sqrt{2}} \right)^L (-1)^{\sum_{r=0}^{L-1} h_{2r+1}(h'_{2r}+h'_{2r+2})} . \tag{10b}$$

This operator can be constructed in terms of local unitary gates and measurements. Namely, one introduces ancillary qubits on the dual lattice and couples them to the original qubits with controlled-Z gates, and then performs projective measurements on the latter [67].

To gain some intuition into the duality defect, consider its action on a product state in the $Z_j$-diagonal basis in $\mathcal{H}_{\text{odd}}$ and $\mathcal{H}_{\text{even}}$ respectively:

$$D_\sigma | \sigma h_1 \sigma h_3 \ldots \sigma h_{2L-1} \rangle = \bigotimes_{r=0}^{L-1} \frac{|0\sigma\rangle + (-1)^{h_{2r-1}+h_{2r+1}} |1\sigma\rangle}{\sqrt{2}}, \tag{11a}$$

$$D_\sigma | h_0 \sigma h_2 \sigma \ldots h_{2L-2} \sigma \rangle = \bigotimes_{r=0}^{L-1} \frac{|\sigma 0\rangle + (-1)^{h_{2r}+h_{2r+2}} |\sigma 1\rangle}{\sqrt{2}}. \tag{11b}$$

Thus for $D_\sigma$ acting on $\mathcal{H}_{\text{odd}}$, the resulting state in $\mathcal{H}_{\text{even}}$ is a product state in the $X$-diagonal basis. Furthermore, for $h_{2r-1}h_{2r+1} = 1$, the output spin at site $2r$ will be parallel to the transverse magnetic field. The action of $D_\sigma$ on $\mathcal{H}_{\text{even}}$ is completely analogous. The operator $D_\sigma$ therefore maps the ground states of the Ising ferromagnet to the ground state of the Ising paramagnet, indeed the action of the Kramers-Wannier duality. Since this is a two-to-one map, the duality defect is necessarily neither invertible nor unitary. More examples illustrating the action of the defects on two qubits have been included in appendix A.

Mapping between Hilbert spaces via the duality defect is the quantum analog of exchanging the lattice of Ising spins/qubits with its dual. It is the reason for our introducing the empty sites labelled by $\sigma$. Moreover, because the duality gates $L_{j,j+1} \otimes \mathbb{1}_{j+2}$ and $\mathbb{1}_j \otimes L_{j+1,j+2}$ from (9) do not commute, $D_\sigma$ cannot be decomposed into a tensor product of local terms. Thus even though $L_{j,j+1}$ is unitary, $D_\sigma$ is not.

The duality and spin-flip defects defined by (9) and (7) are *topological*. They allow the two-dimensional classical Ising model to be defined in the presence of defects, so that the partition function is independent of deformations of the defect paths. The topological behavior arises because these matrix elements obey the *defect commutation relations* [44]

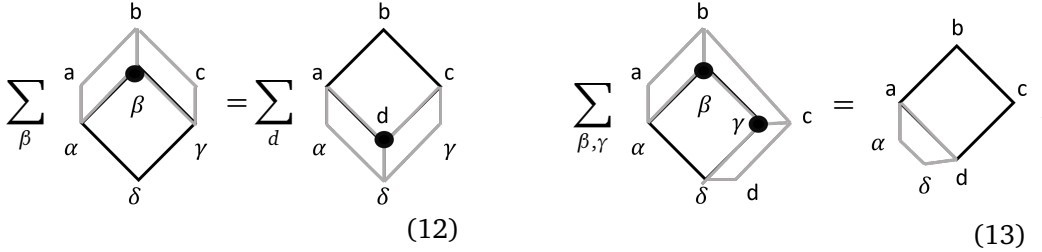

$$\tag{12}$$
$$\tag{13}$$

where the gray quadrilaterals represent either the spin-flip defect or the duality defect. The extension of the proof to the Floquet circuit is immediate, as the defect commutes with each $X_j$ and $Z_{j-1}Z_{j+1}$ individually, and so with each unitary $W_j^X(u_j)$ and $W_j^{ZZ}(u_j)$.

The defects thus can be freely deformed throughout the Floquet circuit, and so $D_\sigma$ and $D_\psi$ commute with the Floquet unitary. Because the duality defect exchanges lattices,

$$D_\sigma U_{\mathbb{1},\text{even}} = U_{\mathbb{1},\text{odd}} D_\sigma \, , \quad D_\sigma U_{\mathbb{1},\text{odd}} = U_{\mathbb{1},\text{even}}D_\sigma, \tag{14}$$

it also toggles between $W^X$ and $W^{ZZ}$. Namely, having $a = c = \sigma$ in (12) or (13) with the duality defect requires $\delta = \beta = \sigma$ as well. The defect commutation relations then relate $W^X$ on the right-hand side to $W^{ZZ}$ on the left-hand side.

It is easy to verify that the two types of topological defects themselves commute with each other: $D_\sigma D_\psi = D_\psi D_\sigma$. A little more work [44] shows that they also obey the *fusion rules*

$$D_\sigma^2 = \mathbb{1} + D_\psi, \qquad D_\sigma D_\psi = D_\psi D_\sigma = D_\sigma, \qquad D_\psi^2 = \mathbb{1}, \tag{15}$$

where $\mathbb{1}$ is the identity operator. Applying these rules gives $D_\sigma^3 = D_\sigma(\mathbb{1} + D_\psi) = 2D_\sigma$. The eigenvalues of $D_\sigma$ are thus $0, \pm\sqrt{2}$. Moreover, because $(D_\psi - \mathbb{1})D_\sigma = 0$, the duality defect projects onto states even under the spin-flip symmetry implemented by $D_\psi$, i.e. $D_\sigma|\omega\rangle = 0$ for any state $|\omega\rangle$ obeying $D_\psi|\omega\rangle = -|\omega\rangle$. The Hilbert space therefore splits into three different sectors $(\sqrt{2}, 1)$, $(-\sqrt{2}, 1)$ and $(0, -1)$, labelled by the eigenvalues of $D_\sigma$ and $D_\psi$ (see Appendix C.1 in particular Table 1 for the continuum theory).

Since the defect operators commute with the unitary $U_{\mathbb{1}}$, these sectors are left invariant by the Floquet evolution. They thus can be thought of as symmetry sectors, even though $D_\sigma$ is not unitary. Kramers-Wannier duality defects thus provide a fundamental implementation of what is now called a *categorical symmetry* or *generalized symmetry* (see [68, 69] for recent reviews). Such topological defects can be found in any two-dimensional classical lattice model or quantum chain built from a fusion category [48, 70]. This mathematical structure underlies the fusion of chiral primary fields in a rational conformal field theory (RCFT) [71] and of anyonic particles [72]. We review the topological defects in the Ising CFT and these structures in Appendix C.1. There is a topological defect associated with each object in the fusion category, with an explicit realization on the lattice, and the defect fusion rules like (15) are those of the corresponding category. In the Ising case, there are three simple objects typically labelled as $\mathbb{1}, \sigma$, and $\psi$, correspondingly the duality and spin-flip defects are labelled as $D_\sigma$ and $D_\psi$, with the identity "defect" $D_{\mathbb{1}} = \mathbb{1}$. We exploit this topological nature of the Ising defects further in section 4 in our definition of twisted boundary conditions.

# 3 Amplitudes in the presence of topological defects

In this section, we study the quantum amplitudes of various states undergoing Floquet evolution with the driven Ising model with topological defects. These amplitudes are essentially the Lorentzian analogues of the partition functions with boundaries in [44] and are the easiest way to observe the effects of defects on Floquet time evolution. These amplitudes can be computed by exact diagonalization. The most straightforward way to compute them is to package the physical qubits and the dual site $|\sigma\rangle$ into qutrits. While this limits the system sizes we can study, small system sizes are sufficient for our purposes of demonstrating the fusion rules of the topological defects in Floquet circuits.

The Floquet unitaries (2a,2b) define the usual Floquet Ising model. We have allowed for arbitrary inhomogenous couplings $u_j$, but we typically restrict them to be one of two distinct values. We call $u_j$ in $W_j^X$ the transverse field $g$, and $u_j$ in $W_j^{ZZ}$ the Ising coupling $J$. Even though there may not be long-range order under Floquet time-evolution, for brevity, the part of the circuit where the Ising coupling $J$ is larger (smaller) than the transverse field coupling $g$ will be referred to as being in the ferromagnetic (paramagnetic) phase. The ferromagnetic and paramagnetic Floquet phases we study are then

$$\text{Paramagnet:} \quad g > J, \quad J + g < \frac{\pi}{2},$$
$$\text{Ferromagnet:} \quad g < J, \quad J + g < \frac{\pi}{2},$$

with both $J, g \geq 0$. When the Ising and transverse field couplings are equal, we refer to the Floquet circuit as being in the critical phase. Moreover, a domain wall refers to the site that separates a paramagnetic phase from a ferromagnetic phase. The condition that the Ising coupling and transverse fields have a sum no greater than $\frac{\pi}{2}$ excludes the $\pi$ phases which are Floquet phases that are unique to the driven system [73–76]. By restricting our attention to the less exotic Floquet phases, the effects of the topological defects on the Floquet Ising model will be easier to tease out. However, in section 5, we briefly discuss $\pi$ modes in the presence of a duality twist.

We start by considering amplitudes in the absence of defects. For simplicity, we take the initial and final states to be one of all qubits fixed up, and all qubits fixed down which in $\mathcal{H}_{\text{even}}$ and $\mathcal{H}_{\text{odd}}$ are:

$$|\text{u}\rangle = |0\sigma 0\sigma \ldots 0\sigma\rangle, \qquad |\text{d}\rangle = |1\sigma 1\sigma \ldots 1\sigma\rangle, \tag{16}$$
$$|\text{u}'\rangle = |\sigma 0\sigma 0 \ldots \sigma 0\rangle, \qquad |\text{d}'\rangle = |\sigma 1\sigma 1 \ldots \sigma 1\rangle. \tag{17}$$

Plots of the real parts of the matrix element $\langle\text{u}|U_{\mathbb{1}}^n|\text{u}\rangle$ in the paramagnetic and ferromagnetic phases are shown in figures 2 and 3 respectively. They oscillate rapidly with a slow beat. To understand the source of the rapid oscillations, consider the exactly solvable limit $u_{2r+1} = 0$ for all $r$. The unitary $U_{\mathbb{1},\text{even}}$ then includes only the transverse field terms, and so is in the paramagnetic phase. Taking the remaining couplings $u_{2r} = u$ to be homogeneous, the matrix element of the defectless Floquet circuit between fixed states is

$$\langle\text{u}|U_{\mathbb{1},\text{even}}^n|\text{u}\rangle = \cos^L nu = \begin{cases} \dfrac{1}{2^{L-1}} \displaystyle\sum_{k=0}^{\frac{L-1}{2}} \binom{L}{k} \cos((L-2k)nu), & L \text{ odd}, \quad (18a) \\[4mm] \dfrac{1}{2^L}\binom{L}{\frac{L}{2}} + \dfrac{1}{2^{L-1}} \displaystyle\sum_{k=0}^{\frac{L}{2}-1} \binom{L}{k} \cos((L-2k)nu), & L \text{ even}. \quad (18b) \end{cases}$$

These amplitudes are superpositions of oscillations with angular frequencies up to $Lu$, and account for the rapid oscillations observed in the matrix elements.

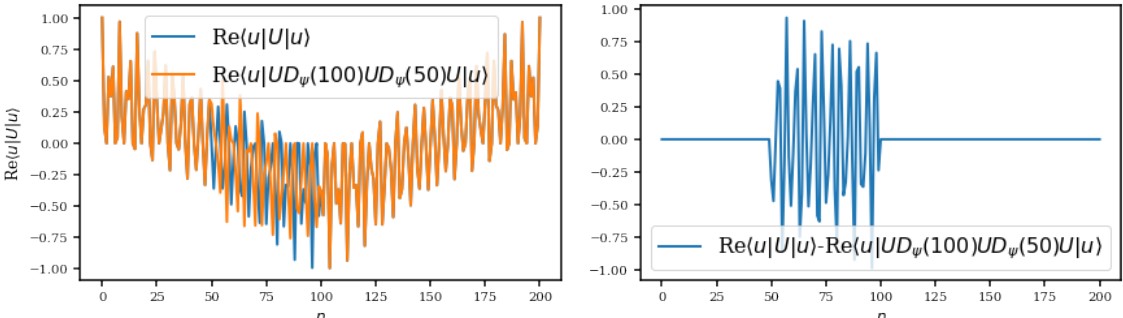

Figure 2: Comparison of amplitudes for a $L = 3$ circuit without defects and with two spin-flip defects inserted at times $n = 50, 100$. The physical spins are on even sites, and the couplings correspond to the paramagnetic phase with transverse magnetic fields $u_{2r} = \frac{\pi}{4}$ and Ising couplings $u_{2r-1} = \frac{\pi}{8}$ for $r = 1, 2, 3$. (Left): Plots of the matrix elements $\text{Re}\langle u|U_{\mathbb{1},\text{even}}|u\rangle$ and the corresponding quantity with spin-flip defects inserted at the $50^{\text{th}}$ and $100^{\text{th}}$ step, $\text{Re}\langle u|U_{\mathbb{1},\text{even}}D_{\psi,\text{even}}(100)U_{\mathbb{1},\text{even}}D_{\psi,\text{even}}(50)U_{\mathbb{1},\text{even}}|u\rangle$, versus the number of Floquet steps $n$. More precisely, spin-flip defects are applied after 49 and 99 defect-less unitaries $U_{\mathbb{1},\text{even}}$ have been applied. (Right): The difference between these two matrix elements.

Since the initial and final states are product states in the $Z$ basis, the matrix element takes an even simpler form in the exactly solvable ferromagnetic phase corresponding to $u_{2r} = 0$. The transverse magnetic fields vanish in the single-step unitary $U_{\mathbb{1},\text{even}}$, leaving only Ising coupling terms. For homogeneous couplings $u_{2r+1} = u$,

$$\langle u|U_{\mathbb{1},\text{even}}^n|u\rangle = e^{-inLu}, \tag{19}$$

so that the amplitude oscillates with a single angular frequency $Lu$ proportional to the total system size. The real part of this amplitude thus vanishes for times $n_m = (m + 1/2)\pi/Lu$ for integer $m$. Plots of this matrix element with a non-zero magnetic field are shown in figure 3. The introduction of non-zero transverse fields lead to an additional oscillation over a much longer time-scale.

## 3.1 Spin-flip defect

The simplest topological defect that can be inserted in the Floquet circuit is the spin-flip defect $D_\psi$. Matrix elements in the paramagnetic phase are compared with the defectless circuit in figure 3. Only the real part of the matrix elements are shown, as the imaginary parts are very similar. Inserting a spin-flip defect noticeably changes the oscillations, but a second insertion makes the amplitude identical to the defectless case. This behavior is a consequence of $D_\psi$ commuting with $U_{\mathbb{1}}$ and squaring to the identity, as seen in (15).

Plots of the return amplitude in the ferromagnetic phase are shown in figure 3. We see that applying $D_\psi$ at about time $n \approx 50$ when the return amplitude $\text{Re}[\langle u|U_{\mathbb{1}}^n|u\rangle]$ is close to zero, causes it to become close to one: $\text{Re}[\langle u|D_\psi U_{\mathbb{1}}^n|u\rangle] \sim 1$. Since $\langle u|D_\psi = \langle d|$, this observation shows that the state $U_{\mathbb{1}}^n|u\rangle \sim |d\rangle$. Thus probing with $D_\psi$ showed that the system evolved close to the all-down state at a certain time. Applying $D_\psi$ again necessarily removes the effects of the first insertion.

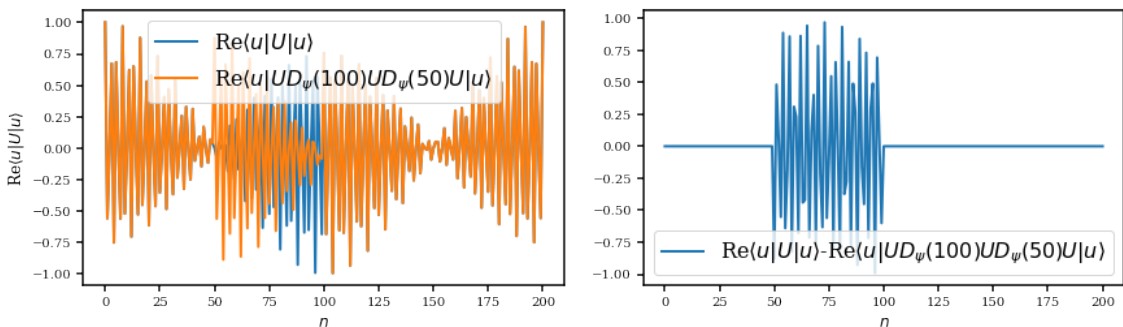

Figure 3: As with figure 2, only in the ferromagnetic phase with transverse magnetic field $u_{2r} = \frac{\pi}{8}$ and Ising couplings $u_{2r-1} = \frac{\pi}{4}$.

## 3.2 Duality defect

We here study the effect of inserting a duality defect $D_\sigma$ into the Floquet circuit. To highlight the fusion rules of the duality and spin-flip defects, we also consider the case when both are present.

Plots of the Floquet circuit with spin-flip and duality defects are shown in figure 4. Unlike the spin flip defect which is unitary and acts exclusively in $\mathcal{H}_{\text{even}}$ or $\mathcal{H}_{\text{odd}}$, the duality defect is not unitary, and connects the two Hilbert spaces. In performing the numerical simulations, the effect of the duality defect has been computed using the transformations (11a), (11b). These are well defined, linear, albeit non-unitary transformations that can be easily implemented once the local Hilbert space is expanded to be a qutrit $(0, 1, \sigma)$.

Since the duality defect interchanges the even and odd sites, to best understand its effect we evaluate the amplitude between two states, one where all spins are up on the even sites $|u\rangle$, and the other where all spins are up on the dual sites i.e., the odd sites $|u'\rangle$. Treating each lattice site as a qutrit $(0, 1, \sigma)$ allows one to expand the Hilbert space to include both even and odd sites. Before the duality defect is applied, the amplitude is zero. After the duality defect is applied, both $\langle u'|$ and $U|u\rangle$ live on the same lattice and the amplitude becomes non-zero. As seen in figure 4, applying a single duality defect at different times leads to the same amplitude after the latest defect has been applied. This is because the duality defect is topological and hence can be moved freely throughout the circuit. Furthermore, applying a single spin-flip defect followed by a duality defect and vice versa gives the same result as applying a single duality defect after the time at which the latest topological defect has been applied. This is because the defects obey the fusion rule (15). Therefore, the spin-flip defect can be fused with the duality defect to produce a single duality defect. Note that these defects can be fused together even though they were applied at different times, and the resulting topological defect is not restricted to a certain time. That is because the defects can be freely moved up and down the circuit since they are topological. Also, the fusion rules (15) were proven by acting the defects on the quantum state one immediately after the other, so they apply even at the shortest time scale of the Floquet circuit.

Plots of the return amplitude for a circuit with two duality defects inserted at two different times are shown in figure 5. This return amplitude is compared with the sum of the return amplitude of a circuit with no defects and a circuit with a single spin-flip defect. Since there are two duality defects, both the initial and final states are chosen to live on the same lattice. During the time between the two duality defects, $U|u\rangle$ and $\langle u|$ live on different lattices so their inner product is zero. After the second duality defect is applied, both states live on the same lattice and have a non-zero inner product. In addition, after the second duality defect is applied, the return amplitude agrees with the sum of the return amplitude of a circuit with no

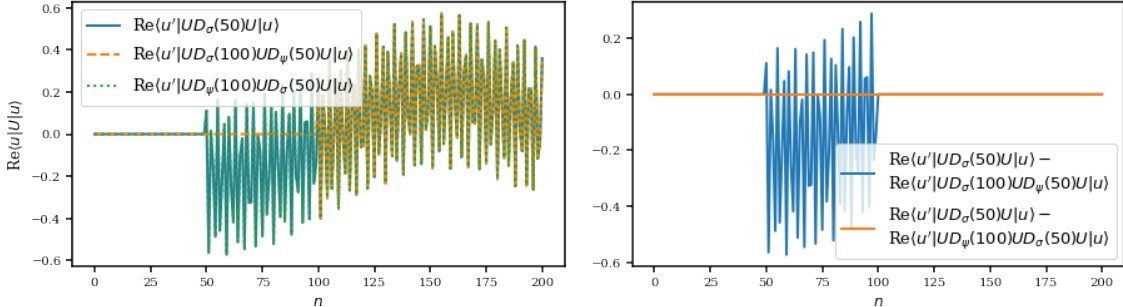

Figure 4: (Left): Plots of the real part of the amplitudes with a single duality defect inserted at $n = 50$, $\mathrm{Re}\langle u'|U_{\mathbb{1},\mathrm{odd}}D_\sigma(50)U_{\mathbb{1},\mathrm{even}}|u\rangle$ (blue), a single spin-flip inserted at $n = 50$ followed by a single duality defect inserted at $n = 100$, $\mathrm{Re}\langle u'|U_{\mathbb{1},\mathrm{odd}}D_\sigma(100)U_{\mathbb{1},\mathrm{even}}D_\psi(50)U_{\mathbb{1},\mathrm{even}}|u\rangle$ (orange) as well as a duality defect at $n = 50$ followed by a spin-flip defect at $n = 100$, $\mathrm{Re}\langle u'|U_{\mathbb{1},\mathrm{odd}}D_\psi(100)U_{\mathbb{1},\mathrm{odd}}D_\sigma(50)U_{\mathbb{1},\mathrm{even}}|u\rangle$ (green), where $U_{\mathbb{1},\mathrm{even}}$ and $U_{\mathbb{1},\mathrm{odd}}$ are the defectless Floquet unitary circuits acting on the even and odd sites respectively. Just as in the previous figures, inserting the defects at the Floquet steps $n = 50$ and $n = 100$ means that the defects are inserted after 49 and 99 defectless unitaries $U_{\mathbb{1}}$ are inserted. Note that these values of $n$ are chosen arbitrarily. The couplings are set to be $u_{2r} = \frac{\pi}{4}$ and $u_{2r+1} = \frac{\pi}{8}$. (Right): The difference between applying only a single duality defect and applying a single spin-flip followed by a single duality defect, $\mathrm{Re}\langle u'|U_{\mathbb{1},\mathrm{odd}}D_\sigma(50)U_{\mathbb{1},\mathrm{even}}|u\rangle - \mathrm{Re}\langle u'|U_{\mathbb{1},\mathrm{odd}}D_\sigma(100)U_{\mathbb{1},\mathrm{even}}D_\psi(50)U_{\mathbb{1},\mathrm{even}}|u\rangle$ (blue), as well as the difference between applying only a single duality defect and applying a single duality defect followed by a single spin-flip defect, $\mathrm{Re}\langle u'|U_{\mathbb{1},\mathrm{odd}}D_\sigma(50)U_{\mathbb{1},\mathrm{even}}|u\rangle - \mathrm{Re}\langle u'|U_{\mathbb{1},\mathrm{odd}}D_\psi(100)U_{\mathbb{1},\mathrm{odd}}D_\sigma(50)U_{\mathbb{1},\mathrm{even}}|u\rangle$ (orange).

defects and a circuit with a single spin-flip defect. This is because the defects obey the fusion rule (15) so introducing two duality defects into the Floquet circuit results in a superposition of matrix elements, one for a defectless circuit and the other for a circuit with a single spin-flip defect.

## 4 Twisted boundary conditions

In the preceding section, we studied the effect on Floquet time evolution of inserting defects stretched across the system i.e, "space-like" defects. The operators $D_\psi$ and $D_\sigma$ implement (categorical) symmetries and so commute with the unitary evolution operator $U_{\mathbb{1}}$ with periodic boundary conditions. In this section, generalizing [44], we utilize the same defect gates to define Floquet evolution in the presence of "time-like" defects which implement *twisted boundary conditions*. Such boundary conditions are special in that the system still obeys a (modified) translation invariance. Since the defects introduce a twist, the corresponding unitaries will be denoted by $T$ instead of $U$.

The spin-flip defect gates (7) allow us to define a unitary operator $T_\psi$ with anti-periodic boundary conditions on the circuit, while the duality defect gates (9) allow us to define a unitary operator $T_\sigma$ with duality-twisted boundary conditions [49–51]. The latter are unusual in that in their presence, part of the circuit can be in the paramagnetic phase and part of it can be in the ferromagnetic phase.

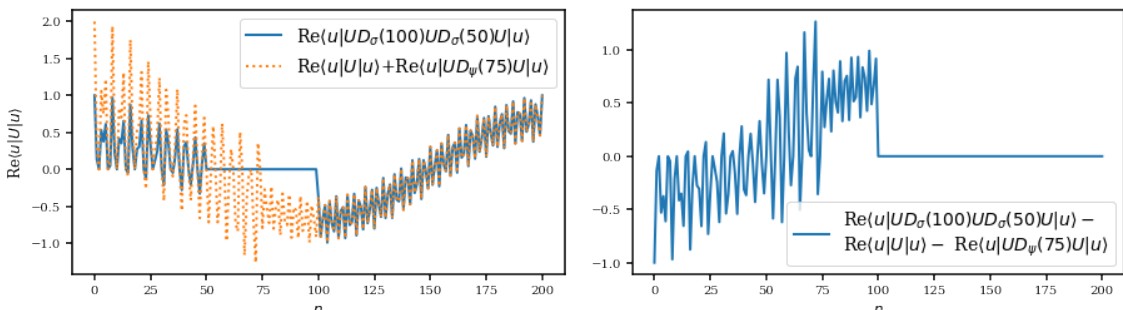

Figure 5: (Left): The real part of the matrix element with two duality defects inserted at $n = 50, 100$ i.e, $\mathrm{Re}\langle u|U_{\mathbb{1},\mathrm{even}}D_{\sigma}(100)U_{\mathbb{1},\mathrm{odd}}D_{\sigma}(50)U_{\mathbb{1},\mathrm{even}}|u\rangle$, where $U_{\mathbb{1},\mathrm{even}}$ and $U_{\mathbb{1},\mathrm{odd}}$ are the defectless Floquet unitary circuits acting on the even and odd sites respectively. Also shown is the sum of the matrix elements for the defectless circuit and a circuit with a single spin-flip defect inserted at $n = 75$, i.e., $\mathrm{Re}\langle u|U_{\mathbb{1},\mathrm{even}}|u\rangle + \mathrm{Re}\langle u|U_{\mathbb{1},\mathrm{even}}D_{\psi}(75)U_{\mathbb{1},\mathrm{even}}|u\rangle$. (Right): Difference between the two curves of the left plot. A total of $L = 3$ physical sites are taken and the couplings are chosen to be $u_{2r} = \frac{\pi}{4}, u_{2r+1} = \frac{\pi}{8}$. Both the initial and final states $|u\rangle$ live on the even sites.

## 4.1 Anti-periodic boundary conditions

Anti-periodic boundary conditions are made using the spin-flip gates (7), but here they must be placed in the vertical "time-like" direction. The matrix elements of a single-step Floquet unitary acting on $\mathcal{H}_{\mathrm{odd}}$ in the presence of a defect at $2s-1$ are then

$$\langle \sigma h'_1 \sigma h'_3 \ldots \sigma h'_{2L-1}|T_{\psi,2s-1,\mathrm{odd}}|\sigma h_1 \sigma h_3 \ldots \sigma h_{2L-1}\rangle$$

$$= \qquad\qquad\qquad\qquad\qquad\qquad\qquad\qquad\qquad\qquad\qquad\qquad \tag{20}$$

As apparent in the picture, an extra qubit is inserted into the system next to that at site $2s-1$, but because of the form of (7), it is fixed to have value $1-h_{2s-1}$. The effect of flipping an Ising spin in $W_j^{ZZ}(u_j)$ is equivalent to sending $u_j \to -u_j$. Since each of the spins $(h_{2s-1}, 1-h_{2s-1})$ is involved in one such weight, the resulting unitaries are

$$T_{\psi,2s-1,\mathrm{even}} = \left(\prod_{r=0}^{L-1} W_{2r}^{X}(u_{2r})\right)\left(\prod_{r=0}^{L-1} W_{2r+1}^{ZZ}(u_{2r+1}(-1)^{\delta_{r,s-1}})\right), \tag{21a}$$

$$T_{\psi,2s-1,\mathrm{odd}} = \left(\prod_{r=0}^{L-1} W_{2r}^{ZZ}(u_{2r}(-1)^{\delta_{r,s}})\right)\left(\prod_{r=0}^{L-1} W_{2r+1}^{X}(u_{2r+1})\right). \tag{21b}$$

Repeatedly acting with $T_{\psi}$ to evolve the system in time carries the defect line along with it. The defect commutation relations ensure that the defect can be moved across the system without changing the physics.

In both even and odd cases, the only effect is to change the sign of a single Ising coupling, which indeed is the usual definition of anti-periodic boundary conditions. Thus in the trivially

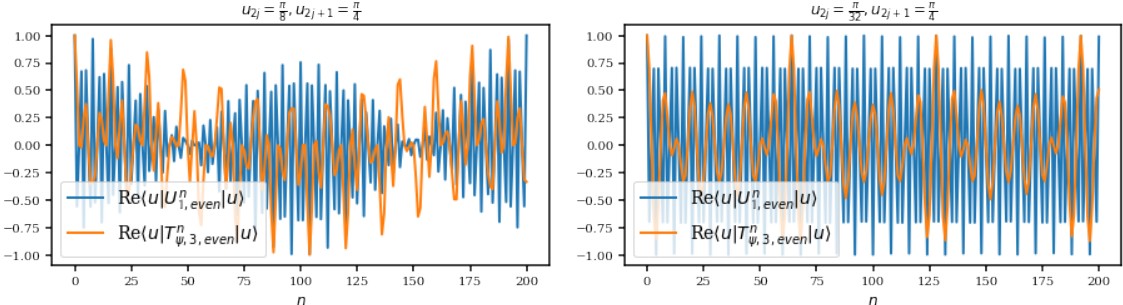

Figure 6: Comparing evolution with periodic (defectless) and anti-periodic boundary conditions in the ferromagnetic phase. $L = 3$ physical qubits live on the even sites and the anti-periodic boundary condition is imposed at site 3. Plots of $\mathrm{Re}\langle\mathrm{u}|U^n_{\mathbb{1},\mathrm{even}}|\mathrm{u}\rangle$ and $\mathrm{Re}\langle\mathrm{u}|T^n_{\psi,3,\mathrm{even}}|\mathrm{u}\rangle$ with $u_{2r} = \frac{\pi}{8}, u_{2r-1} = \frac{\pi}{4}$ (left) and $u_{2r} = \frac{\pi}{32}, u_{2r-1} = \frac{\pi}{4}$ (right). The closer one is to the solvable ferromagnetic phase at $u_{2r} = 0$, the more enhanced is the suppression of the oscillation frequency due to the spin-flip defect.

solvable paramagnetic case with only transverse magnetic fields, anti-periodicity has no effect. However, in the fully ferromagnetic limit, the anti-periodicity modifies the return amplitude. Namely, setting $u_{2r} = 0$ in (21a) leaves only Ising couplings, and keeping the latter to be uniform, $u_{2r-1} = u$, gives

$$\langle\mathrm{u}|T^n_{\psi,2s-1,\mathrm{even}}|\mathrm{u}\rangle = e^{-in(L-2)u}. \tag{22}$$

Compared with the same matrix element for the defectless circuit (19), the angular frequency has been reduced because of the twist. This effect will not be noticeable in the thermodynamic limit since $L-2$ would be effectively the same as $L$.

The ferromagnetic phase being more interesting, plots of a circuit with anti-periodic boundary conditions are shown in figure 6 for two different couplings, both in the ferromagnetic phase. The effect of the anti-periodicity is most apparent when approaching the trivially solvable point of $u_{2r} = 0$. In particular, when the transverse magnetic field is small, the oscillations are less erratic. The introduction of the anti-periodic boundary condition reduces the frequency of oscillation since the sign of an Ising coupling term is flipped and the resulting cancellation reduces the oscillation frequency.

## 4.2 Duality-twisted boundary conditions

Among the different configuration of defects, the duality-twisted boundary conditions is particularly interesting and forms the main subject of this sub-section and the next section. The single duality defect gates in (9) are the building blocks of the Floquet unitary with duality-twisted boundary conditions. In order to construct this unitary, note that, just as the spin-flip defect gate, the duality defect gate (9) can be inserted vertically in the Floquet circuit to implement duality-twisted boundary conditions. Since the Floquet unitary is applied repeatedly, we rearrange the terms to make the expression tractable. For simplicity, we introduce the twist between the ends of the chain, i.e, sites $2L-1$ and 0. The single step unitary time evolution

with the twist, and acting on $\mathcal{H}_{\text{even}}$ and $\mathcal{H}_{\text{odd}}$ have the matrix elements

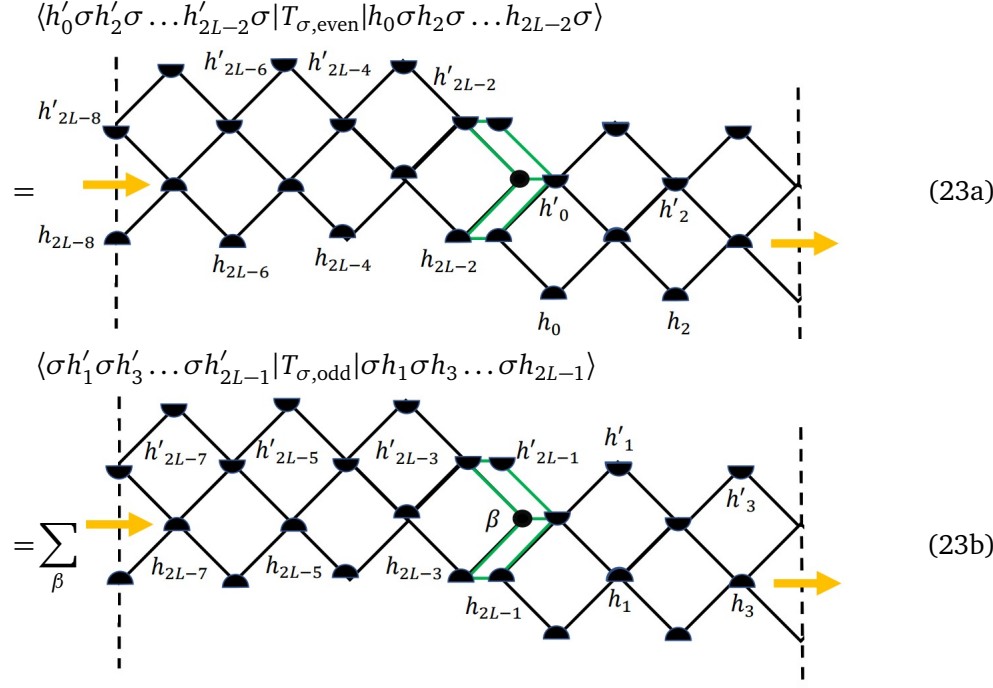

Above, the yellow arrows indicate the edges that are identified as a result of the periodic boundary conditions, with $L = 6$ in the above diagrams. It is easy to see that there are only $2L - 1$ couplings $u_i$ instead of $2L$, one less than the defectless circuit or the circuit with the anti-periodic boundary conditions.

We would now like to write down explicit expressions for the green quadrilaterals in (23). In order to do so, note that, depending on the orientation of the duality defect gate, it can be represented either as a single site Hadamard gate or a two-site controlled-$Z$ gate. For example, if the two physical sites of a gate are stacked directly on top of each other (as in (23b)), this duality defect gate will correspond to a single-site Hadamard gate

$$\frac{1}{\sqrt{2}}(-1)^{h_j h'_{j+1}} = H_{h_j, h'_{j+1}}, \qquad \frac{1}{\sqrt{2}}(-1)^{h'_j h_{j+1}} = H_{h'_j, h_{j+1}}. \tag{24}$$

The Hadamard gate is a single qubit quantum logic gate that exchanges the eigenvectors of the Pauli $X$ and Pauli $Z$ matrices with the eigenvalue 1, as well as the eigenvectors of the Pauli $X$ and Pauli $Z$ matrices with the eigenvalue $-1$. If, on the other hand, the two physical sites are horizontally aligned (as in (23a)), then the duality defect gate corresponds to a two-qubit gate whose matrix element is precisely that of the controlled-$Z$ gate

$$\frac{1}{\sqrt{2}}(-1)^{h_j h'_{j+1}} = \frac{1}{\sqrt{2}} CZ_{h_j h'_{j+1}, h_j h'_{j+1}}, \qquad \frac{1}{\sqrt{2}}(-1)^{h'_j h_{j+1}} = \frac{1}{\sqrt{2}} CZ_{h'_j h_{j+1}, h'_j h_{j+1}}. \tag{25}$$

The controlled-$Z$ gate is a two qubit quantum logic gate that does nothing to a product of eigenstates in the $Z$ basis unless both of them are in the spin down state in which case the controlled-$Z$ gate produces an overall factor of $-1$. It is interesting to note that both the Hadamard gate and controlled-$Z$ gates are Clifford gates which are a special subset of unitaries that map a single string of Pauli matrices to a single string of Pauli matrices [77, 78].

We denote $\tilde{L}_j$ as the duality defect gate normalized by the inclusion of the quantum dimensions (4), which from the above discussion, takes the form

$$\tilde{L}_j = \quad\vcenter{\hbox{}}_{j-1 \ j \ \ j+1} \quad = \quad \vcenter{\hbox{}}_{j-1 \ j \ \ j+1}$$

$$= |2\rangle\langle 2|_{j-1} \otimes H_j \otimes |2\rangle\langle 2|_{j+1} + CZ_{j-1,j+1} \otimes |2\rangle\langle 2|_j. \tag{26}$$

The normalization factors coming from the duality defect gates (9) as well as the quantum dimensions (4) combine to render the normalized Floquet unitary with the duality-twisted boundary conditions, unitary, even though, as we have seen, the duality defect in the space-like direction is not unitary. Thus, the explicit expressions of the single step Floquet unitary that acts in $\mathcal{H}_{\text{even}}$ and $\mathcal{H}_{\text{odd}}$ are

$$T_{\sigma,\text{even}} = \left(\prod_{j=0}^{L-2} W_{2j+1}^{ZZ}(u_{2j+1})\right) CZ_{2L-2,0} W_{2L-2}^{X}(u_{2L-2}) CZ_{2L-2,0} \left(\prod_{j=0}^{L-2} W_{2j}^{X}(u_{2j})\right), \tag{27a}$$

$$T_{\sigma,\text{odd}} = \left(\prod_{j=0}^{L-2} W_{2j+1}^{X}(u_{2j+1})\right) H_{2L-1} W_{2L-2}^{ZZ}(u_{2L-2}) H_{2L-1} \left(\prod_{j=0}^{L-2} W_{2j}^{ZZ}(u_{2j})\right). \tag{27b}$$

Conjugating a transverse field unitary with a controlled-$Z$ gate or an Ising coupling unitary with a Hadamard gate produces the mixed coupling unitaries $CZ_{2L-2,0} e^{-iu_{2L-2}X_{2L-2}} CZ_{2L-2,0} = e^{-iu_{2L-2}X_{2L-2}Z_0}$ and $H_{2L-1} e^{-iu_{2L-2}Z_{2L-3}Z_{2L-1}} H_{2L-1} = e^{-iu_{2L-2}Z_{2L-3}X_{2L-1}}$ respectively. Therefore, the Floquet unitaries in the presence of the duality-twisted boundary conditions can alternatively be written as

$$T_{\sigma,\text{even}} = \left(\prod_{j=0}^{L-2} W_{2j+1}^{ZZ}(u_{2j+1})\right) e^{-iu_{2L-2}X_{2L-2}Z_0} \left(\prod_{j=0}^{L-2} W_{2j}^{X}(u_{2j})\right), \tag{28a}$$

$$T_{\sigma,\text{odd}} = \left(\prod_{j=0}^{L-2} W_{2j+1}^{X}(u_{2j+1})\right) e^{-iu_{2L-2}Z_{2L-3}X_{2L-1}} \left(\prod_{j=0}^{L-2} W_{2j}^{ZZ}(u_{2j})\right). \tag{28b}$$

The net effect of introducing duality-twisted boundary conditions is to convert a transverse field unitary (when even sites are physical sites) and an Ising coupling unitary (when odd sites are physical sites) into mixed coupling terms of the $XZ$ kind. More precisely, when the physical spins live on the even sites, the duality-twist removes the Ising coupling between site $2L-2$ and $0$. Instead, the transverse magnetic field term at site $2L-2$ gets converted into a mixed coupling term between sites $2L-2$ and $0$. On the other hand, when the physical spins live on the odd sites, introducing a duality-twist removes the transverse field term at site $2L-1$ and converts the Ising coupling between the sites $2L-3$ and $2L-1$ into a mixed coupling term. In either case, there is one less transverse magnetic field and there is a single mixed coupling term.

Since $H_{2L-1}$ does not commute with $W_0^{ZZ}(u_0)$ in (27b), and $CZ_{2L-2,0}$ does not commute with $W_0^{X}(u_0)$ in (27a), these local unitary transformations cannot convert the defectless unitaries into duality twisted unitaries.

It was shown in [44, 48] that the Hamiltonian for the Ising model with duality-twisted boundary conditions has a degenerate eigenspectrum. Let us extend this argument to the Floquet unitary with the same boundary conditions. The Floquet unitary can be written in terms of a Floquet Hamiltonian $H_F$ as $T_{\sigma,\text{odd}} = e^{-iH_F}$. In general $H_F$ is long-ranged in the

couplings, but in the small coupling (high frequency) limit it takes the following local form in $\mathcal{H}_{\text{odd}}$

$$H_F = \sum_{j=0}^{L-2} u_{2j+1} X_{2j+1} + u_{2L-2} Z_{2L-3} X_{2L-1} + \sum_{j=0}^{L-2} u_{2j} Z_{2j-1} Z_{2j+1}. \tag{29}$$

We define the Floquet unitary that is transformed by a phase gate at site $2L-1$ as $\tilde{T}_{\sigma,\text{odd}} := S_{2L-1} T_{\sigma,\text{odd}} S_{2L-1}^\dagger := e^{-i\tilde{H}_F}$. $S$ is the phase gate

$$S = \begin{pmatrix} 1 & 0 \\ 0 & i \end{pmatrix}, \tag{30}$$

which is a single qubit gate that exchanges the Pauli $X$ and $Y$ matrices up to a sign. The corresponding Floquet Hamiltonian is

$$\tilde{H}_F = \sum_{j=0}^{L-2} u_{2j+1} X_{2j+1} + u_{2L-2} Z_{2L-3} Y_{2L-1} + \sum_{j=0}^{L-2} u_{2j} Z_{2j-1} Z_{2j+1}. \tag{31}$$

Since $[\tilde{H}_F, D_\psi] = 0$, the Floquet unitary and the spin-flip operators are simultaneously diagonalizable. We denote such a simultaneous eigenstate by $|\varepsilon, \omega\rangle$ where

$$\tilde{T}_{\sigma,\text{odd}} |\varepsilon, \omega\rangle = e^{-i\varepsilon} |\varepsilon, \omega\rangle, \qquad D_\psi |\varepsilon, \omega\rangle = \omega |\varepsilon, \omega\rangle, \tag{32}$$

with $\omega = \pm 1$. One can show that

$$D_\psi Z_{2L-1} |\varepsilon, \omega\rangle^* = -\omega Z_{2L-1} |\varepsilon, \omega\rangle^*, \tag{33}$$

since the spin-flip operator and its eigenvalues are real, and also because this operator anti-commutes with $Z_{2L-1}$. One can also show that

$$e^{i\tilde{H}_F} Z_{2L-1} |\varepsilon, \omega\rangle^* = e^{i\varepsilon} Z_{2L-1} |\varepsilon, \omega\rangle^*. \tag{34}$$

To derive the above equation note that taking the complex conjugate of $\tilde{H}_F$ keeps all terms unchanged except the mixed-coupling term $Z_{2L-3} Y_{2L-1}$. Conjugating $e^{i\tilde{H}_F^*}$ by $Z_{2L-1}$ flips the sign of $Y_{2L-1}$, and nothing else, leading to $\left( Z_{2L-1} e^{-i\tilde{H}_F} Z_{2L-1} \right)^* = e^{i\tilde{H}_F}$. Now, bringing the Floquet unitary and the phase in (34) to opposite sides of the equation gives

$$e^{-i\tilde{H}_F} Z_{2L-1} |\varepsilon, \omega\rangle^* = e^{-i\varepsilon} Z_{2L-1} |\varepsilon, \omega\rangle^*, \tag{35}$$

showing that the eigenspectrum is degenerate with the degenerate pairs being $|\varepsilon, \omega\rangle, Z_{2L-1} |\varepsilon, \omega\rangle^*$. The above manipulations do not generalize away from the high frequency limit because $\left( Z_{2L-1} \tilde{T}_{\sigma,\text{odd}} Z_{2L-1} \right)^* \neq \tilde{T}_{\sigma,\text{odd}}^\dagger$ in general. Inspecting the spectrum obtained from exact diagonalization also shows that the degeneracy does not hold away from the small coupling/high frequency limit. Thus we conclude that the degeneracy of the eigenspectrum does not hold for the Floquet unitary apart from the high frequency limit.

In the remaining paper, we will further explore the properties of the Floquet unitary with duality-twisted boundary conditions.

## 5 Majorana zero mode

When open boundary conditions are imposed, Floquet SPT phases are known to host edge modes [73, 79–81]. In the absence of the twist, and with periodic boundary conditions, the

system hosts no isolated localized modes. We will show below that the duality-twisted boundary conditions allow for the introduction of a single domain wall in the circuit separating a paramagnetic and a ferromagnetic phase. We will analyze this Floquet circuit and show that a single unpaired Majorana zero mode resides on the domain wall. Following this, we will show that even in the absence of a domain wall, a Majorana zero mode exists, but it is localized where the duality twisted boundary condition has been imposed. In addition, we will show that at the critical point where all couplings are equal, the Majorana zero mode is delocalized.

We will present an explicit construction of the Majorana zero mode with duality-twisted boundary conditions. We will also detect the Majorana fermion by the appropriate auto-correlation functions. These direct methods unambiguously establish the existence of a Majorana zero mode. Majorana modes in Floquet SPT phases have so far been studied for open boundary conditions, where the Majorana modes appear in pairs. In addition, these Majorana modes are also detectable via non-zero winding numbers. Exploring the circuits with twisted boundary conditions from the viewpoint of SPT phases, and constructing associated winding numbers, will be left for future work.

## 5.1 Majorana zero mode in the presence of a domain wall

The circuit with duality-twisted boundary conditions, (28), is capable of hosting a single domain wall. In particular, for a homogeneous choice of couplings, a domain wall located at $2s-1$ is obtained by setting $u_0 = \ldots = u_{2s-2} = u_{2s-1} = \ldots = u_{2L-3} = u'$ to be identical and $u_{2s} = \ldots = u_{2L-2} = u_1 = \ldots = u_{2s-3} = u$ to be identical. Then, the single step unitary with duality-twisted boundary conditions and the domain wall becomes

$$T_{\sigma,\text{even}} = \left(\prod_{j=0}^{s-2} W_{2j+1}^{ZZ}(u)\right)\left(\prod_{j=s-1}^{L-2} W_{2j+1}^{ZZ}(u')\right) e^{-iuX_{2L-2}Z_0}\left(\prod_{j=0}^{s-1} W_{2j}^{X}(u')\right)\left(\prod_{j=s}^{L-2} W_{2j}^{X}(u)\right), \quad (36a)$$

$$T_{\sigma,\text{odd}} = \left(\prod_{j=0}^{s-2} W_{2j+1}^{X}(u)\right)\left(\prod_{j=s-1}^{L-2} W_{2j+1}^{X}(u')\right) e^{-iuZ_{2L-3}X_{2L-1}}\left(\prod_{j=0}^{s-1} W_{2j}^{ZZ}(u')\right)\left(\prod_{j=s}^{L-2} W_{2j}^{ZZ}(u)\right). \quad (36b)$$

The above assumes that $2 \leq s \leq L-2$.

We now show how a localized Majorana zero mode can appear. The symmetry operator in the presence of duality-twisted boundary conditions is no longer $D_\psi$ but $\Omega$, which for physical spins on the odd sites is

$$\Omega = iZ_{2L-1}D_\psi = -(X_1 \ldots X_{2L-3})Y_{2L-1} = S_{2L-1}^\dagger D_\psi S_{2L-1}. \quad (37)$$

By performing a Jordan-Wigner transformation, the Floquet unitary can be written in terms of fermionic operators. Following Appendix A of [44], we define the Jordan-Wigner transformation to be

$$\gamma_{2j} = \left(\prod_{k=-1}^{j-1} X_{2k+1}\right)Z_{2j+1}, \qquad \gamma_{2j+1} = -\left(\prod_{k=-1}^{j-1} X_{2k+1}\right)Y_{2j+1}, \qquad j = 0, \ldots, L-2, \quad (38)$$

which consists of Pauli strings starting from the mixed coupling term at site $2L-1$. The Majorana fermions at that site are defined similarly but without the string of Pauli $X$ matrices attached, $\gamma_{2L-2} = Z_{2L-1}$, $\gamma_{2L-1} = -Y_{2L-1}$. These Majorana fermions satisfy the usual anti-commutation relations $\{\gamma_{2j}, \gamma_{2k}\} = \{\gamma_{2j+1}, \gamma_{2k+1}\} = 2\delta_{kl}$ and $\{\gamma_{2j}, \gamma_{2k+1}\} = 0$. With these definitions, all the Majorana fermions can be shown to commute with the symmetry operator (37) except for $\gamma_{2L-2}$ which anti-commutes with it,

$$\Omega\gamma_{2L-2} = -\gamma_{2L-2}\Omega, \qquad \Omega\gamma_{2j} = \gamma_{2j}\Omega, \quad j = 0, \ldots, L-2 \quad (39a)$$

$$\Omega\gamma_{2j+1} = \gamma_{2j+1}\Omega, \quad j = 0, \ldots, L-1. \quad (39b)$$

Applying the Jordan-Wigner transformation (38), the transverse field and Ising coupling terms in the Floquet unitary become

$$Z_{2j-1}Z_{2j+1} = -i\gamma_{2j-1}\gamma_{2j}, \qquad j = 0, \dots, L-2, \tag{40a}$$

$$X_{2j+1} = -i\gamma_{2j}\gamma_{2j+1}, \qquad j = 0, \dots, L-1, \tag{40b}$$

$$Z_{2L-3}X_{2L-1} = -i\Omega\gamma_{2L-3}\gamma_{2L-1}. \tag{40c}$$

Therefore, the Floquet unitary in (28b) can be written as a product of exponents of quadratic fermion terms

$$T_{\sigma,\text{odd}} = e^{-\sum_{j=0}^{L-2} u_{2j+1}\gamma_{2j}\gamma_{2j+1}} e^{-u_{2L-2}\Omega\gamma_{2L-3}\gamma_{2L-1}} e^{-\sum_{j=0}^{L-2} u_{2j}\gamma_{2j-1}\gamma_{2j}}, \tag{41}$$

where the Majorana fermion $\gamma_{2L-2}$ is noticeably absent in the Floquet unitary. Despite this, $\gamma_{2L-2}$ does not commute with the Floquet unitary because of the presence of the symmetry operator $\Omega$ attached to the mixed coupling term, and because $\Omega$ anti-commutes with $\gamma_{2L-2}$.

Under the Floquet unitary with duality-twisted boundary conditions, the Majorana fermions evolve as follows

$$
\begin{aligned}
T_{\sigma,\text{odd}}^\dagger \gamma_{2j} T_{\sigma,\text{odd}} =\ & \gamma_{2j}\cos 2u_{2j+1}\cos 2u_{2j} + \gamma_{2j-1}\sin 2u_{2j}\cos 2u_{2j+1} \\
& -\gamma_{2j+1}\sin 2u_{2j+1}\cos 2u_{2j+2} + \gamma_{2j+2}\sin 2u_{2j+1}\sin 2u_{2j+2}, \quad j = 0, \dots, L-3,
\end{aligned}
\tag{42a}
$$

$$
\begin{aligned}
T_{\sigma,\text{odd}}^\dagger \gamma_{2j+1} T_{\sigma,\text{odd}} =\ & \gamma_{2j+1}\cos 2u_{2j+1}\cos 2u_{2j+2} + \gamma_{2j}\sin 2u_{2j+1}\cos 2u_{2j} \\
& -\gamma_{2j+2}\sin 2u_{2j+2}\cos 2u_{2j+1} + \gamma_{2j-1}\sin 2u_{2j+1}\sin 2u_{2j}, \quad j = 0, \dots, L-3,
\end{aligned}
\tag{42b}
$$

$$
\begin{aligned}
T_{\sigma,\text{odd}}^\dagger \gamma_{2L-4} T_{\sigma,\text{odd}} =\ & \gamma_{2L-4}\cos 2u_{2L-3}\cos 2u_{2L-4} + \gamma_{2L-5}\sin 2u_{2L-4}\cos 2u_{2L-3} \\
& -\gamma_{2L-3}\sin 2u_{2L-3}\cos 2u_{2L-2} + \Omega\gamma_{2L-1}\sin 2u_{2L-3}\sin 2u_{2L-2}\cos 2u_0 \\
& -\Omega\gamma_0\sin 2u_{2L-3}\sin 2u_{2L-2}\sin 2u_0,
\end{aligned}
\tag{42c}
$$

$$
\begin{aligned}
T_{\sigma,\text{odd}}^\dagger \gamma_{2L-3} T_{\sigma,\text{odd}} =\ & \gamma_{2L-3}\cos 2u_{2L-3}\cos 2u_{2L-2} - \Omega\gamma_{2L-1}\sin 2u_{2L-2}\cos 2u_{2L-3}\cos 2u_0 \\
& +\Omega\gamma_0\sin 2u_{2L-2}\cos 2u_{2L-3}\sin 2u_0 + \gamma_{2L-4}\sin 2u_{2L-3}\cos 2u_{2L-4} \\
& +\gamma_{2L-5}\sin 2u_{2L-3}\sin 2u_{2L-4},
\end{aligned}
\tag{42d}
$$

$$
T_{\sigma,\text{odd}}^\dagger \gamma_{2L-1} T_{\sigma,\text{odd}} = \gamma_{2L-1}\cos 2u_0\cos 2u_{2L-2} - \gamma_0\sin 2u_0\cos 2u_{2L-2} + \Omega\gamma_{2L-3}\sin 2u_{2L-2},
\tag{42e}
$$

$$
\begin{aligned}
T_{\sigma,\text{odd}}^\dagger \gamma_{2L-2} T_{\sigma,\text{odd}} =\ & \cos 2u_{2L-2}\gamma_{2L-2} + \Omega\sin 2u_{2L-2}\cos 2u_0\gamma_{2L-2}\gamma_{2L-3}\gamma_{2L-1} \\
& -\Omega\sin 2u_{2L-2}\sin 2u_0\gamma_{2L-2}\gamma_{2L-3}\gamma_0.
\end{aligned}
\tag{42f}
$$

Since the Majorana fermion $\gamma_{2L-2}$ does not appear in the Floquet unitary (41), the other Majorana fermions will not transform into operators that contain it. Organizing the remaining $2L-1$ Majorana fermions into a vector $\vec{\gamma} = (\gamma_0, \gamma_1, \dots, \gamma_{2L-3}, \gamma_{2L-1})$, these Majorana fermions will transform under a single Floquet step under the action of an orthogonal matrix

$$T_{\sigma,\text{odd}}^\dagger \vec{\gamma} T_{\sigma,\text{odd}} = M\vec{\gamma}, \tag{43}$$

where $M$ is an orthogonal matrix. Diagonalizing the matrix $M = VDV^{-1}$ such that $D_{ij} = \delta_{ij}\lambda_i$, and defining $\Psi_i = V_{ik}^{-1}\gamma_k$, the new linear combination of Majorana fermions evolve as

$$T_{\sigma,\text{odd}}^\dagger \Psi_i T_{\sigma,\text{odd}} = D_{il}\Psi_l = \lambda_i\Psi_i. \tag{44}$$

If there is an eigenvalue of 1, the corresponding linear combination of Majorana fermions is left invariant under the Floquet evolution and therefore commutes with the Floquet unitary.[1] Note that these eigenvectors correspond to linear combinations of all the Majorana fermions except $\gamma_{2L-2}$ and hence commute with the symmetry operator $\Omega$ by (39). Therefore, the Majorana fermion that commutes with the Floquet unitary is a Majorana zero mode but not a strong mode in the sense of [55, 57, 82–90] where the Majorana strong mode anti-commutes with the discrete symmetry.

We now introduce a domain wall in the Floquet unitary by setting $u_0 = \ldots = u_{2s-2} = u_{2s-1} = \ldots = u_{2L-3} = u'$ to be identical and $u_{2s} = \ldots = u_{2L-2} = u_1 = \ldots = u_{2s-3} = u$ to be identical. As a simple example, consider a system with $L = 6$ sites and a domain wall located at $2s - 1 = 5$. Defining $c = \cos 2u, c' = \cos 2u', s = \sin 2u, s' = \sin 2u'$, the explict form of the matrix $M$ is

$$
M = \begin{pmatrix}
cc' & -sc' & ss' & 0 & 0 & 0 & 0 & 0 & 0 & 0 & cs' \\
sc' & cc' & -cs' & 0 & 0 & 0 & 0 & 0 & 0 & 0 & ss' \\
0 & cs' & cc' & -sc' & ss' & 0 & 0 & 0 & 0 & 0 & 0 \\
0 & ss' & sc' & cc' & -cs' & 0 & 0 & 0 & 0 & 0 & 0 \\
0 & 0 & 0 & c's' & c'^2 & -cs' & ss' & 0 & 0 & 0 & 0 \\
0 & 0 & 0 & s'^2 & c's' & cc' & -sc' & 0 & 0 & 0 & 0 \\
0 & 0 & 0 & 0 & 0 & sc' & cc' & -cs' & ss' & 0 & 0 \\
0 & 0 & 0 & 0 & 0 & ss' & cs' & cc' & -sc' & 0 & 0 \\
-\Omega ss'^2 & 0 & 0 & 0 & 0 & 0 & 0 & sc' & cc' & -cs' & \Omega ss'c' \\
\Omega ss'c' & 0 & 0 & 0 & 0 & 0 & 0 & ss' & cs' & cc' & -\Omega sc'^2 \\
-cs' & 0 & 0 & 0 & 0 & 0 & 0 & 0 & 0 & \Omega s & cc'
\end{pmatrix}. \tag{45}
$$

The third and fourth rows are basically repetitions of the first and second, but shifted to the right by two columns while respecting periodic boundary conditions. This pattern does not carry through to the remaining rows because of the presence of a domain wall at $2s - 1 = 5$, with the fifth and sixth rows now encapsulating the domain wall. The seventh and eighth rows can be obtained from the third and fourth rows by exchanging the primed and unprimed quantities because the couplings $u$ and $u'$ exchange roles after passing through the domain wall. The last three rows encapsulate the duality-twisted boundary condition. With this structure, it is straightforward to generalize $M$ to a system of arbitrary length as the bulk (such as the first and second rows and the seventh and eighth rows) will keep repeating until either the domain wall or the duality-twisted boundary is encountered.

We now solve for the Majorana zero mode in some exactly solvable limits before we give the general solution. With the above choice of couplings, the Floquet unitary with the duality twist becomes

$$
T_{\sigma,\text{odd}} = e^{-\sum_{j=0}^{s-2} u\gamma_{2j}\gamma_{2j+1} - \sum_{j=s-1}^{L-2} u'\gamma_{2j}\gamma_{2j+1}} e^{-u\Omega\gamma_{2L-3}\gamma_{2L-1}} e^{-\sum_{j=0}^{s-1} u'\gamma_{2j-1}\gamma_{2j} - \sum_{j=s}^{L-2} u\gamma_{2j-1}\gamma_{2j}}. \tag{46}
$$

When $u' = 0$, the Majorana fermion $\gamma_{2s-2}$ does not appear in the Floquet unitary, and therefore it commutes with the Floquet unitary and the Majorana zero mode is simply $\Psi(u' = 0) = \gamma_{2s-2}$. This corresponds to the single unpaired Majorana fermion at the location of the domain wall as shown on the left of figure 7. On the other hand, when $u = 0$, all the Majorana fermions except $\gamma_{2L-2}$ appear. As seen on the right of figure 7, all the Majorana fermions are paired up except for the three Majorana fermions located in the vicinity of the domain wall. Under the

---

[1]In Appendix C.3, we give an explanation of this feature in the continuum using the Ising fusion category (see figure 20).



Figure 7: Pairing of Majorana fermions for the odd lattice sites when one of the couplings is set to zero. The diagram on the left corresponds to $u' = 0$ while the diagram on the right corresponds to $u = 0$. In the paramagnetic part of the circuit, on-site Majorana fermions are paired up while in the ferromagnetic part of the circuit, the even Majorana fermions of each site are paired up with the odd Majorana fermions of an adjacent site.

Floquet unitary, these three Majorana fermions transform as

$$T_{\sigma,\text{odd}}^\dagger \begin{pmatrix} \gamma_{2s-1} \\ \gamma_{2s-2} \\ \gamma_{2s-3} \end{pmatrix} T_{\sigma,\text{odd}} = \begin{pmatrix} \cos 2u' & \sin 2u' \cos 2u' & \sin^2 2u' \\ -\sin 2u' & \cos^2 2u' & \sin 2u' \cos 2u' \\ 0 & -\sin 2u' & \cos 2u' \end{pmatrix} \begin{pmatrix} \gamma_{2s-1} \\ \gamma_{2s-2} \\ \gamma_{2s-3} \end{pmatrix}. \tag{47}$$

There is an eigenvector $(1, -\tan u', 1)/\sqrt{1 + \sec^2 u'}$ with eigenvalue 1, so that the Majorana fermion that commutes with the Floquet unitary is

$$\Psi(u = 0) = \frac{1}{\sqrt{1 + \sec^2 u'}} (\gamma_{2s-1} - \tan u' \gamma_{2s-2} + \gamma_{2s-3}). \tag{48}$$

In other words, the three Majorana fermions located in the vicinity of the domain wall can be rotated so that two of them will be paired up, leaving behind a single unpaired Majorana fermion.

The Floquet unitary also simplifies when either of the couplings is set to $\frac{\pi}{2}$. If we set $u' = \frac{\pi}{2}$, the Floquet unitary becomes

$$T_{\sigma,\text{odd}}(u' = \frac{\pi}{2}) = (-1)^L e^{-u \sum_{j=0}^{s-2} \gamma_{2j} \gamma_{2j+1}} \left( \prod_{j=2s-1}^{2L-3} \gamma_j \right) e^{-u\Omega \gamma_{2L-3} \gamma_{2L-1}} \left( \prod_{j=2L-1}^{2s-3} \gamma_j \right) e^{-u \sum_{j=s}^{L-2} \gamma_{2j-1} \gamma_{2j}}. \tag{49}$$

Since $\gamma_{2s-2}$ is absent in the Floquet unitary, it will be a Majorana zero mode. If we now set $u = \pi/2$, the Floquet unitary becomes

$$T_{\sigma,\text{odd}}(u = \frac{\pi}{2})$$
$$= (-1)^{L-2} \left( \prod_{j=0}^{s-2} \gamma_{2j} \gamma_{2j+1} \right) e^{-\sum_{j=s-1}^{L-2} u' \gamma_{2j} \gamma_{2j+1}} e^{-u\Omega \gamma_{2L-3} \gamma_{2L-1}} e^{-\sum_{j=0}^{s-1} u' \gamma_{2j-1} \gamma_{2j}} \left( \prod_{j=s}^{L-2} \gamma_{2j-1} \gamma_{2j} \right). \tag{50}$$

All the Majorana fermions are present, just as in the $u = 0$ case. The three Majorana fermions $\gamma_{2s-1}$, $\gamma_{2s-2}$ and $\gamma_{2s-3}$ rotate amongst themselves according to

$$T_{\sigma,\text{odd}}^\dagger \begin{pmatrix} \gamma_{2s-1} \\ \gamma_{2s-2} \\ \gamma_{2s-3} \end{pmatrix} T_{\sigma,\text{odd}} = \begin{pmatrix} -\cos 2u' & \sin 2u' \cos 2u' & \sin^2 2u' \\ \sin 2u' & \cos^2 2u' & \sin 2u' \cos 2u' \\ 0 & \sin 2u' & -\cos 2u' \end{pmatrix} \begin{pmatrix} \gamma_{2s-1} \\ \gamma_{2s-2} \\ \gamma_{2s-3} \end{pmatrix}. \tag{51}$$

The above matrix on the r.h.s. is orthogonal and has an eigenvalue of 1 with an eigenvector of $(1, \cot u', 1)$. Therefore, it has a zero mode given by

$$\Psi(u = \frac{\pi}{2}) = \frac{1}{\sqrt{1 + \csc^2 u'}} (\gamma_{2s-1} + \cot u' \gamma_{2s-2} + \gamma_{2s-3}). \tag{52}$$

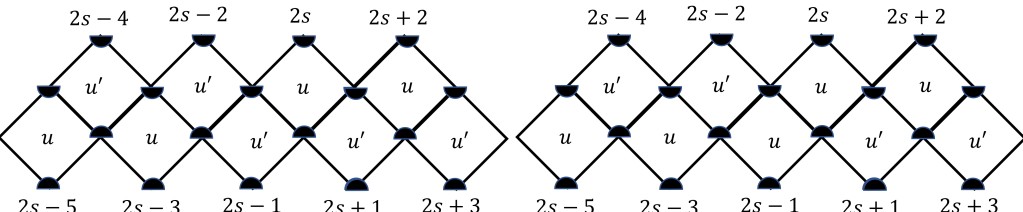

Figure 8: Two possible kinds of domain walls depending upon whether the site $2s-1$ is included with the phase on the right of the domain wall (left plot) or the phase on the left of the domain wall (right plot).

These limits around $u = 0$ ($u = \pi/2$) and $u' = 0$ ($u' = \pi/2$) suggest the existence of a Majorana zero mode at the domain wall for arbitrary values of $u$ and $u'$. In the Kitaev chain with open boundary conditions, there are an even number of Majorana fermions. Depending on the parameters, the Majorana fermions can be paired up intrasite or intersite, leaving either no unpaired Majorana fermions or two unpaired Majorana fermions on the boundary respectively. In our Floquet circuit, introducing a single duality-twisted boundary removes one Majorana fermion which effectively reduces the system size to $L - \frac{1}{2}$. Since there are now an odd number of Majorana fermions, there should always be a single unpaired Majorana zero mode sitting on the domain wall.

Note that there are two different ways of introducing a domain wall at $2s - 1$. Either $u_{2s-1} = u'$ which is the example discussed so far. Alternately, one may set $u_{2s-1} = u$. These two different domain walls are shown in figure 8. In particular, the sites to the left of $2s - 1$ are in a different phase from the sites to the right of $2s - 1$ since the couplings $u$ and $u'$ play opposite roles on opposite sides of the domain wall at $2s - 1$. The site at $2s - 1$ can be included with the phase on the right which corresponds to the choice made in this section, or it can be included with the phase on the left. This will lead to an asymmetry in the couplings $u$ and $u'$ which can be attributed to the asymmetry about the domain wall. To understand this point further, note that for the alternate domain wall, the Floquet unitary is

$$T_{\sigma,\text{odd}} = e^{-\sum_{j=0}^{s-1} u\gamma_{2j}\gamma_{2j+1} - \sum_{j=s}^{L-2} u'\gamma_{2j}\gamma_{2j+1}} e^{-u\Omega\gamma_{2L-3}\gamma_{2L-1}} e^{-\sum_{j=0}^{s-1} u'\gamma_{2j-1}\gamma_{2j} - \sum_{j=s}^{L-2} u\gamma_{2j-1}\gamma_{2j}}. \tag{53}$$

As before, $\gamma_{2L-2}$, which is one of the Majorana fermions located at the duality-twisted boundary, is absent. Now, however, when $u = 0$, the Majorana fermion $\gamma_{2s-1}$ is absent and hence commutes with the Floquet unitary. This is opposite to what we found earlier with our original choice of domain walls.

Having established the presence of a Majorana zero mode at the domain wall in certain simple limits, let us now show that the Majorana zero mode is present for more general couplings $u$ and $u'$. This can be done by computing the eigenvector $\Psi$ of the orthogonal matrix $M$ with unit eigenvalue. The eigenvalue equation $(M - \mathbb{1})\Psi = 0$ translates into the following recursion relations

$$\begin{pmatrix} \Psi_{2j+1} \\ \Psi_{2j+2} \end{pmatrix} = C(u, u') \begin{pmatrix} \Psi_{2j-1} \\ \Psi_{2j} \end{pmatrix}, \qquad j = 0, \dots, s-2, \tag{54a}$$

$$\begin{pmatrix} \Psi_{2s-1} \\ \Psi_{2s} \end{pmatrix} = \begin{pmatrix} 1 & 0 \\ \cot 2u - \frac{\cos 2u'}{\sin 2u} & \frac{\sin 2u'}{\sin 2u} \end{pmatrix} \begin{pmatrix} \Psi_{2s-3} \\ \Psi_{2s-2} \end{pmatrix}, \tag{54b}$$

$$\begin{pmatrix} \Psi_{2j+1} \\ \Psi_{2j+2} \end{pmatrix} = C(u', u) \begin{pmatrix} \Psi_{2j-1} \\ \Psi_{2j} \end{pmatrix}, \qquad j = s, \dots, L-3, \tag{54c}$$

where we defined the matrix

$$C(u,u') = \frac{1}{\sin 2u \sin 2u'} \begin{pmatrix} \sin^2 2u' & \sin 2u'(\cos 2u' - \cos 2u) \\ \sin 2u'(\cos 2u' - \cos 2u) & 1 - 2\cos 2u \cos 2u' + \cos^2 2u' \end{pmatrix}, \quad (55)$$

and assumed that $\sin 2u$ is non-zero. These are a set of recurrence relations that begin with the Majorana fermions at the ends of the chain $\Psi_{2L-1}$ and $\Psi_0$. From (54b), we immediately see that $\Psi_{2s-1} = \Psi_{2s-3}$. Components of the Majorana fermion that commutes with the Floquet unitary can be written as powers of the matrix $C$ by repeated application of the recursion relation

$$\begin{pmatrix} \Psi_{2j+1} \\ \Psi_{2j+2} \end{pmatrix} = C(u,u')^{j+1} \begin{pmatrix} \Psi_{2L-1} \\ \Psi_0 \end{pmatrix}, \qquad j = 0, \dots, s-2, \quad (56a)$$

$$\begin{pmatrix} \Psi_{2j+1} \\ \Psi_{2j+2} \end{pmatrix} = C(u',u)^{j-s+1} \begin{pmatrix} \Psi_{2s-1} \\ \Psi_{2s} \end{pmatrix}, \qquad j = s, \dots, L-3. \quad (56b)$$

The exchange of the arguments $u$ and $u'$ in the second row reflects the presence of a domain wall at the site $2s - 1$. The matrix $C(u,u')$ has eigenvalues $\epsilon_{u,u'}^{\pm 1}$ where

$$\epsilon_{u,u'} = \frac{\tan u}{\tan u'}, \quad (57)$$

and the corresponding eigenvectors are $(\sin u', \cos u')$ and $(-\cos u', \sin u')$ respectively. Therefore, for positive integers $n \in \mathbb{N}$, the $n^{\text{th}}$ power of the matrix is

$$C(u,u')^n = \begin{pmatrix} \epsilon_{u,u'}^n \sin^2 u' + \epsilon_{u,u'}^{-n} \cos^2 u' & (\epsilon_{u,u'}^n - \epsilon_{u,u'}^{-n}) \sin u' \cos u' \\ (\epsilon_{u,u'}^n - \epsilon_{u,u'}^{-n}) \sin u' \cos u' & \epsilon_{u,u'}^n \cos^2 u' + \epsilon_{u,u'}^{-n} \sin^2 u' \end{pmatrix}. \quad (58)$$

This matrix has unit determinant. The coefficients of the eigenvector with unit eigenvalue that lie after the domain wall are related to the Majorana fermions at the ends of the chain by

$$\begin{pmatrix} \Psi_{2j+1} \\ \Psi_{2j+2} \end{pmatrix} = \begin{pmatrix} a_j & b_j \\ c_j & d_j \end{pmatrix} \begin{pmatrix} \Psi_{2L-1} \\ \Psi_0 \end{pmatrix}, \qquad j = s, \dots, L-3, \quad (59)$$

where the entries of the matrix are

$$a_j = \epsilon_{u,u'}^{2+j-2s} \cos^2 u' + \epsilon_{u,u'}^{2s-2-j} \sin^2 u', \quad (60a)$$

$$b_j = (\epsilon_{u,u'}^{2s-2-j} - \epsilon_{u,u'}^{2+j-2s}) \cos u' \sin u', \quad (60b)$$

$$c_j = \epsilon_{u,u'}^{2s-2-j} \cot u \sin^2 u' - \epsilon_{u,u'}^{2-2s+j} \cos^2 u' \tan u, \quad (60c)$$

$$d_j = \cos u' \sin u' (\epsilon_{u,u'}^{2s-2-j} \cot u + \epsilon_{u,u'}^{2+j-2s} \tan u). \quad (60d)$$

The last coefficients that can be obtained from these recursion relations are $\Psi_{2L-5}$ and $\Psi_{2L-4}$ and can thus be written in terms of $\Psi_{2L-1}$ and $\Psi_0$. The remaining equations for the Floquet evolution of the Majorana fermions $\gamma_{2L-4}, \gamma_{2L-3}$ and $\gamma_{2L-1}$ lead to the following system of linear equations

$$0 = (\cos 2u' \cos 2u - 1)\Psi_{2L-4} + \sin 2u \cos 2u' \Psi_{2L-5} - \sin 2u' \cos 2u \Psi_{2L-3}$$
$$+ \Omega \Psi_{2L-1} \sin 2u' \sin 2u \cos 2u' - \Omega \Psi_0 \sin^2 2u' \sin 2u, \quad (61a)$$

$$0 = (\cos 2u' \cos 2u - 1)\Psi_{2L-3} - \Omega \sin 2u \cos^2 2u' \Psi_{2L-1}$$
$$+ \Omega \sin 2u \cos 2u' \sin 2u' \Psi_0 + \sin 2u' \cos 2u \Psi_{2L-4} + \sin 2u' \sin 2u \Psi_{2L-5}, \quad (61b)$$

$$0 = (\cos 2u' \cos 2u - 1)\Psi_{2L-1} - \sin 2u' \cos 2u \Psi_0 + \Omega \sin 2u \Psi_{2L-3}. \quad (61c)$$

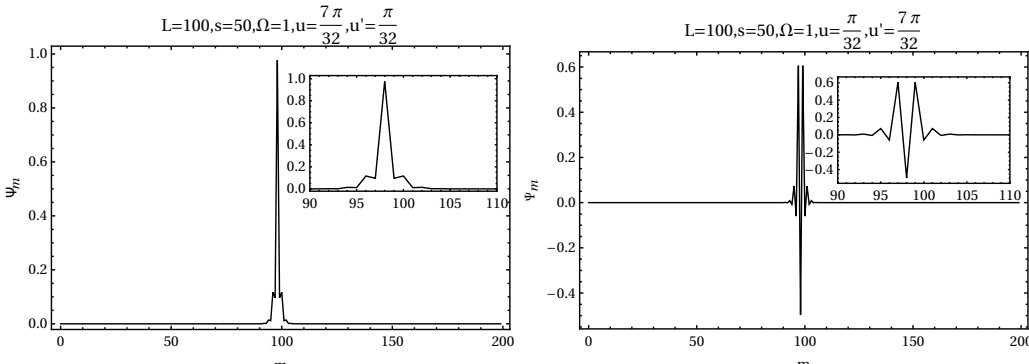

Figure 9: Plots of the analytic solution for the Majorana zero mode $\Psi_m$ against the Majorana mode number $m$ where $m = 0, 1, \ldots, 2L-3, 2L-1$. We take a total system size of $L = 100$ with the domain wall located at $2s-1 = 99$. The plot on the left is for $(u, u') = (\frac{7\pi}{32}, \frac{\pi}{32})$ while the plot on the right is for $(u, u') = (\frac{\pi}{32}, \frac{7\pi}{32})$. The insets show a close-up view of the localized Majorana zero mode.

Since the eigenvector can only be determined up to an overall constant, the last coefficient can be set to $\Psi_{2L-1} = 1$. Then, (61c) gives

$$\Psi_{2L-3} = \frac{1 - \cos 2u' \cos 2u + \sin 2u' \cos 2u \Psi_0}{\Omega \sin 2u}. \tag{62}$$

Substituting this into (61b) and using (59) to relate $\Psi_{2L-5}$ and $\Psi_{2L-4}$ to the coefficients $\Psi_{2L-1}$ and $\Psi_0$, we obtain

$$\Psi_0 = \frac{1 + \cos^2 2u' - 2\cos 2u \cos 2u' - \Omega \sin 2u \sin 2u'(c_{L-3}\cos 2u + a_{L-3}\sin 2u)}{\sin 2u'(\cos 2u' - \cos 2u) + \Omega \sin 2u' \sin 2u(d_{L-3}\cos 2u + b_{L-3}\sin 2u)}. \tag{63}$$

This solution for $\Psi_0$ and $\Psi_{2L-1} = 1$ also satisfies the remaining equation (61a) and thus constitutes a solution to the full system of linear equations given by $(M - \mathbb{1})\Psi = 0$. The remaining coefficients can be obtained by multiplying $(\Psi_{2L-1}, \Psi_0)$ by the appropriate matrices as in (56a) and (59).

Plots of the normalized analytic solution of the Majorana zero mode $\Psi$ are shown in figure 9. The coefficients of the Majorana zero mode is negligible except for the Majorana fermions that are located close to the domain wall at $2s-1$. When $u > u'$, there is a maximum at $\gamma_{2s-2}$ which corresponds to a string of Pauli $X$ matrices with a Pauli $Z$ operator at $2s-1$, the location of the domain wall. Recall that when $u' = 0$, the Majorana zero mode is exactly $\Psi(u' = 0) = \gamma_{2s-2}$. The exact closed-form solution shows that the Majorana zero mode remains localized at the domain wall even after turning on the coupling $u'$. On the other hand, when $u < u'$, the Majorana fermions with the largest coefficients are $\gamma_{2s-3}$, $\gamma_{2s-1}$ and $\gamma_{2s-2}$ with the coefficient of $\gamma_{2s-2}$ having a sign that is opposite to that of the other two. This agrees perfectly with the Majorana zero mode obtained earlier for the $u = 0$ limit (48). In summary, the exact closed-form expression for the Majorana zero mode is consistent with the $u = 0$ and $u' = 0$ limits discussed earlier in the section, and shows that the physical picture of a single unpaired Majorana zero mode at the domain wall extends to non-zero couplings $u$ and $u'$.

From the plots, the Majorana zero mode appears to be symmetric about $2s-2$. To see that this is indeed the case, note that the other coefficients can be related to $\Psi_{2s-2}$ and $\Psi_{2s-3} = \Psi_{2s-1}$

by powers of the inverse of $C(u, u')$ (see Appendix B for the proof)

$$
\begin{pmatrix} \Psi_{2j-1} \\ \Psi_{2j} \end{pmatrix} = C(u,u')^{-(s-1-j)} \begin{pmatrix} \Psi_{2s-3} \\ \Psi_{2s-2} \end{pmatrix}, \qquad j = 0, \dots, s-2, \tag{64a}
$$

$$
\begin{pmatrix} \Psi_{2j+1} \\ \Psi_{2j} \end{pmatrix} = C(u,u')^{-(j-s+1)} \begin{pmatrix} \Psi_{2s-1} \\ \Psi_{2s-2} \end{pmatrix}, \qquad j = s, \dots, L-3, \tag{64b}
$$

where the second row for (64b) holds for $j = L - 2$ as well. This is because (56b) and (54b) allows us to relate $\Psi_{2L-4}$ to $\Psi_{2s-1}$ and $\Psi_{2s-2}$ as $\Psi_{2L-4} = (\epsilon_{u,u'}^{-(L-1-s)} - \epsilon_{u,u'}^{L-1-s}) \sin u' \cos u' \Psi_{2s-1} + (\epsilon_{u,u'}^{L-1-s} \sin^2 u' + \epsilon_{u,u'}^{-(L-1-s)} \cos^2 u') \Psi_{2s-2}$.

We now derive an asymptotic approximation for the Majorana zero mode for the case where the domain wall is located far away from the duality-twisted boundary i.e, $L \gg s \gg 1$.[2] By symmetry, we can focus our attention on the components of the Majorana mode to the left of $\Psi_{2s-2}$. Since (56a) relates both $\Psi_{2j+1}$ and $\Psi_{2j+2}$ to $\Psi_0$ separately, these relations can be combined to relate $\Psi_{2j+1}$ and $\Psi_{2j+2}$ giving

$$
\Psi_{2j+1} = \frac{(\epsilon_{u,u'}^{j+1} - \epsilon_{u,u'}^{-1-j}) \sin u' \cos u' \Psi_{2j+2} + 1}{\epsilon_{u,u'}^{j+1} \cos^2 u' + \epsilon_{u,u'}^{-j-1} \sin^2 u'}, \qquad j = 0, \dots, s-2. \tag{65}
$$

On the other hand, the first recurrence relation (54a) gives

$$
\Psi_{2j+1} = \frac{\sin 2u'}{\sin 2u} \Psi_{2j-1} + \frac{\cos 2u' - \cos 2u}{\sin 2u} \Psi_{2j}, \qquad j = 0, \dots, s-2. \tag{66}
$$

When $\frac{\pi}{2} > u > u' > 0$, $\epsilon_{u,u'} > 1$ so for $1 \ll j \le s-2$, (65) relates odd-numbered fermions to the next even fermion as

$$
\Psi_{2j+1} \approx \tan u' \Psi_{2j+2}, \qquad 1 \ll j \le s-2. \tag{67}
$$

Substituting this into (66) relates each even-numbered fermion with the next fermion as

$$
\Psi_{2j} \approx \cot u \, \Psi_{2j+1}, \qquad 1 \ll j \le s-2. \tag{68}
$$

These two equations can be repeatedly applied to give

$$
\Psi_{2j+1} \approx \tan u' \epsilon_{u,u'}^{j-s+2} \Psi_{2s-2}, \qquad \Psi_{2j} \approx \epsilon_{u,u'}^{j+1-s} \Psi_{2s-2}, \qquad 0 < u' < u < \frac{\pi}{2}, 1 \ll j \le s-2. \tag{69}
$$

Since $\epsilon_{u,u'} > 1$, the coefficients get smaller the further the Majorana fermion is from $\Psi_{2s-2}$.

On the other hand, when $\frac{\pi}{2} > u' > u > 0$, and $\epsilon_{u,u'} < 1$, (65) is approximately

$$
\Psi_{2j+1} \approx -\cot u' \Psi_{2j+2}, \qquad 1 \ll j \le s-2. \tag{70}
$$

Substituting this into (66) gives

$$
\Psi_{2j} \approx -\tan u \, \Psi_{2j+1}, \qquad 1 \ll j \le s-2. \tag{71}
$$

By repeatedly applying these two relations, the other Majorana fermions to the left of $\Psi_{2s-2}$ can be related to it by

$$
\Psi_{2j+1} \approx -\cot u' \epsilon_{u,u'}^{s-2-j} \Psi_{2s-2}, \qquad \Psi_{2j} \approx \epsilon_{u,u'}^{s-1-j} \Psi_{2s-2}, \qquad 0 < u < u' < \frac{\pi}{2}, 1 \ll j \le s-2. \tag{72}
$$

---

[2]See Appendix C.3 for discussions of the domain wall Majorana zero mode in the continuum Ising field theory.

Since $\epsilon_{u,u'} < 1$, the coefficient of the Majorana modes decreases as one moves away from $\Psi_{2s-2}$. Furthermore, the odd coefficients have a sign that is opposite to that of the even coefficients which explains the oscillating sign observed in figure 9.

We now make contact with Floquet SPTs with open boundary conditions, and in particular those that can host zero and $\pi$ Majorana modes with the latter oscillating at twice the drive period [57,89]. The presence and absence of the zero and $\pi$ mode can be inferred from the properties of the orthogonal matrix $M$. Since $M$ is real, for each eigenvector $\vec{v}$ with eigenvalue $\lambda$, $M\vec{v} = \lambda\vec{v}$ we have $M\vec{v}^* = \lambda^*\vec{v}^*$, so that the eigenvalues will come in complex conjugate pairs. This only hinges on $M$ being real and not on its orthogonality. Since $M$ is an orthogonal matrix, its determinant is $\pm 1$. For the case we are studying, the matrix $M$ is composed of $SO(2)$ orthogonal rotations generated by transverse field, mixed coupling and Ising coupling unitaries. Therefore, the determinant of $M$ is 1. Since $M$ has an odd dimension, the complex conjugate pairing will inevitably lead to a single unpaired eigenvalue which must be $\pm 1$. The products of the eigenvalues with complex conjugate pairs is 1, so this last unpaired eigenvalue must be 1, implying that there is always a zero mode. The odd dimension of $M$ always ensures that there is a zero mode even in the absence of a domain wall (see below for an explicit construction). Moreover, when all the couplings are equal, i.e., in the critical phase, this zero mode is delocalized.

We now briefly discuss $\pi$ Majorana modes. Since the determinant of $M$ is 1, $\pi$ modes will have to come in pairs so that the product of their eigenvalues is 1. However, we only have a single "boundary" and so our boundary conditions do not allow for a pair of $\pi$ modes.

## 5.2 Auto-correlation functions

In situations where an analytic solution is not available, one can still identify zero modes by studying auto-correlation functions. In this section, we compute the auto-correlation function defined by

$$A_{\mathcal{O}}(n) = \frac{1}{2^L}\mathrm{Tr}\left[\mathcal{O}(n)\mathcal{O}\right], \tag{73}$$

where $\mathcal{O}$ is an arbitrary operator and $\mathcal{O}(n)$ is the operator after $n$ steps of the Floquet evolution in the Heisenberg picture. In particular, we are interested in computing the auto-correlation function of the Majorana fermions with even mode numbers $\gamma_{2i}$ for $i = 0, \ldots, L-1$. These are the Majorana fermions that correspond to strings of Pauli $X$ operators that begin at the duality-twisted boundary at site $2L - 1$ and end with a Pauli $Z$ operator at site $2i + 1$. We are primarily interested in the auto-correlation function in the presence of duality-twisted boundary conditions as the auto-correlation function may be able to detect the zero mode that is localized at the domain wall.

As in the previous section, we consider the case where the qubits live on the odd integer sites. Plots of the real part of the auto-correlation function for the Majorana fermions $\gamma_{2i}$ as functions of the Floquet period for a system with $L = 9$ sites are shown in figure 10. Various choices of the couplings $u$ and $u'$ are chosen, while the positions of the domain wall is fixed at $2s - 1 = 7$, and the position of the duality-twisted boundary is fixed at the end of the chain as in (23b). The imaginary parts of the auto-correlation functions are not shown since they vanish. Four different locations of $\gamma_{j-1}$ are chosen, of which two of them, $j = 2L - 2, 0$, are adjacent to the duality-twisted boundary, while the other two, $j = 2s - 2, 2s$, are adjacent to the domain wall. When $u > u'$, the auto-correlation function of the Majorana fermion $\gamma_{2s-2}$ oscillates about a non-zero value that decreases as $u'$ approaches $u$, i.e. as the system approaches the critical point. This non-zero value is given by the square of the coefficient of $\gamma_{2s-2}$ in the expansion of the Majorana zero mode $\Psi$ and is indicated by a black horizontal line in figure 10. Once the system is at the critical point, none of the auto-correlation functions appear to oscillate about a non-zero value. This is because the domain wall vanishes at the

critical point since $u = u'$, and no localized Majorana mode is expected. On the other side of the critical point where $u' > u$, the auto-correlation function for the Majorana mode $\gamma_{2i}$ appears to oscillate rapidly about a non-zero value, and this oscillation is not as clean as for the $u > u'$ case. This is not surprising as the Majorana zero mode in the $u = 0$ limit is given by a linear combination of $\gamma_{2s-1}$, $\gamma_{2s-3}$ and $\gamma_{2s-2}$ rather than $\gamma_{2s-2}$ alone as in the $u' = 0$ case. Nevertheless, the auto-correlation function of $\gamma_{2s-2}$ oscillates about a non-zero value that agrees very well with the analytic solution for the amplitude-squared of the $\gamma_{2s-2}$ fermion in the exact solution of the Majorana mode (solid black line).

**Vary Couplings**

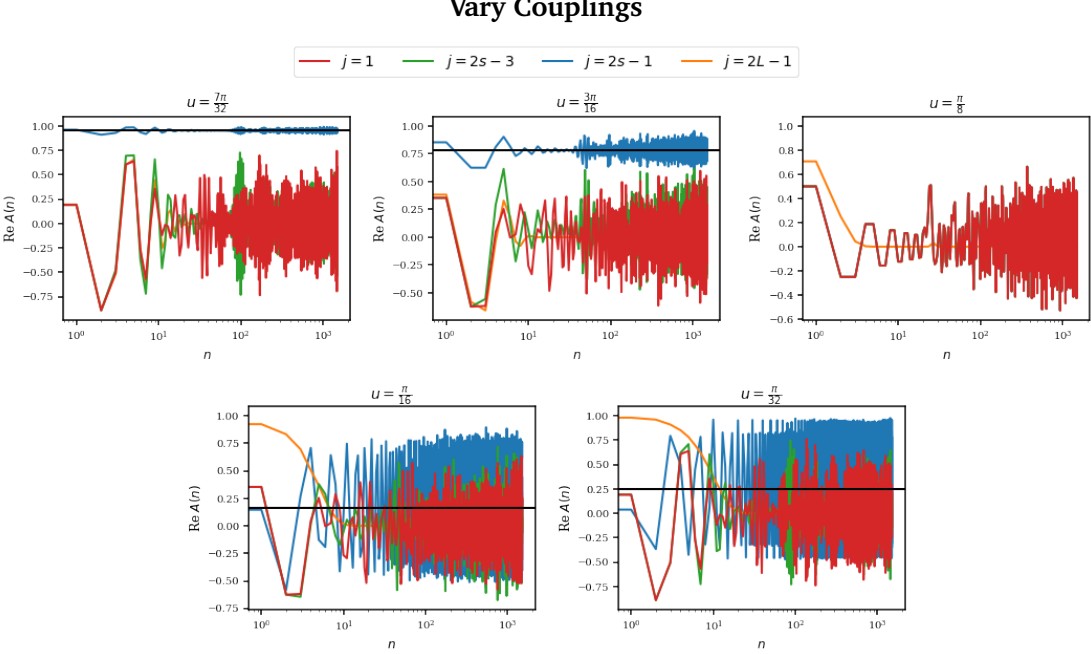

Figure 10: Plots of the real parts of the auto-correlation function $A(n)$ as functions of Floquet periods as the position of the Majorana fermion $\gamma_{j-1}$ at the odd site $j$ is varied across the chain. The total system size is $L = 9$, with the duality-twisted boundary condition imposed at site $2L - 1$ as in (23b), and the domain wall is fixed at $2s - 1 = 7$. Each subplot corresponds to a single set of couplings but for different positions of the Majorana fermion $\gamma_{j-1}$. The couplings $u$ and $u'$ are tuned across a phase transition while keeping $u + u' = \frac{\pi}{4}$. Because of the domain wall, part of the circuit is paramagnetic while the other part of the circuit is ferromagnetic except for the third subplot ($u = u' = \pi/8$) which is at the critical point. More precisely, for the first two plots of the upper panel, the sites $j = 1, \ldots, 2s - 3$ correspond to the paramagnetic part of the circuit while the sites $j = 2s - 1, \ldots, 2L - 3$ correspond to the ferromagnetic part of the circuit since $u > u'$. On the other hand, for the lower plots since $u' > u$, the sites $j = 1, \ldots, 2s - 3$ are in the ferromagnetic phase while the sites $j = 2s - 1, \ldots, 2L - 3$ are in the paramagnetic phase. The amplitude squared of the $\gamma_{2s-2}$ fermion in the Majorana zero mode $\Psi$ is indicated by the black horizontal line for $L = 100$ which is large enough to approximate the thermodynamic limit.

Plots of the real part of the auto-correlation function of the even Majorana fermion $\gamma_{2i}$, for different positions of the domain wall are shown in figure 11. As the domain wall is moved along the periodic chain, the auto-correlation function that oscillates about a non-zero value, follows the domain wall. This is because the Majorana zero mode is localized at the domain wall. Furthermore, the auto-correlation function does not oscillate about a non-zero value at any other location other than in the vicinity of the domain wall.

**Vary Domain Wall**

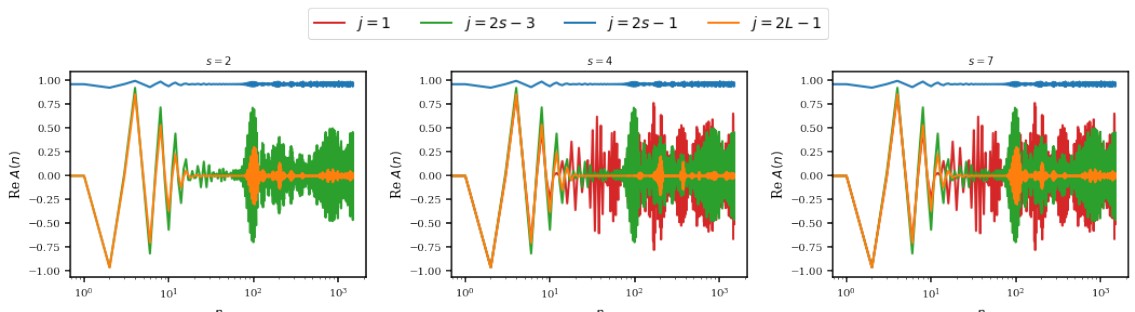

Figure 11: Same as in figure 10 except that the couplings are fixed to be $u = \frac{\pi}{4}$ and $u' = \frac{\pi}{32}$, while the domain wall position is varied according to $2s - 1 = 3, 7, 13$ (left, middle and right plots respectively). The locations $j = 1, \ldots, 2s - 3$ correspond to the paramagnetic part of the circuit while the sites $j = 2s - 1, \ldots, 2L - 3$ correspond to the ferromagnetic part of the circuit since $u > u'$. $L = 9$ as in figure 10.

Plots of the auto-correlation functions for the even Majorana fermion $\gamma_{2i}$ as the total system size is varied, are shown in figure 12. At early times, the auto-correlation functions do not appear to depend on the total system size $L$ and the plots for different system sizes lie on top of each other. After a certain time, the auto-correlation functions for different total system sizes are no longer identical with deviations appearing at later times for larger total system sizes. This deviation can thus be attributed to finite size effects. Therefore, the auto-correlation function is expected to oscillate about a non-zero value in the thermodynamic limit for the Majorana fermion $\gamma_{2s-2}$ (when $u > u'$) and about zero for the even Majorana fermions located at other sites. It is also worth noting that the auto-correlation function shows no discernible dependence on the parity of the total system size. The dependence of the Majorana zero mode on the total system size differs from that of the strong zero mode [55, 57, 82–88, 90] because the Majorana zero mode studied here is exactly conserved and can thus be thought of as a separate $\mathbb{Z}_2$ symmetry of the system.

### 5.3 Majorana zero mode in the absence of the domain wall

In this section we will demonstrate that with duality-twisted boundary conditions, a Majorana zero mode exists even in the absence of a domain wall, with the mode localized at the duality-twisted boundary.

Consider the case where the physical spins are placed on the odd sites. We take $u_{2i+1} = g$ and $u_{2i} = J$ which are the transverse field and Ising couplings respectively. From (41) the Floquet unitary is given by

$$T_\sigma = e^{-g \sum_{j=0}^{L-2} \gamma_{2j} \gamma_{2j+1}} e^{-J\Omega \gamma_{2L-3} \gamma_{2L-1}} e^{-J \sum_{j=0}^{L-2} \gamma_{2j-1} \gamma_{2j}}. \tag{74}$$

When $J = 0$ this is $T_\sigma(J = 0) = \prod_{j=0}^{L-2} e^{-igX_{2j+1}}$ which is simply a product of transverse fields acting on sites 1 to $2L - 3$. For this case, both $\gamma_{2L-1}$ and $\gamma_{2L-2}$ commute with the unitary because there is no term acting on site $2L - 1$. Thus in this simple limit, there is a zero mode, $\gamma_{2L-1}$, located at the duality-twisted boundary.

In contrast, when $g = 0$, all the Majorana fermions except for $\gamma_{2L-2}$ are present in the Floquet unitary. In addition, from (42), the Floquet unitary rotates the following three Majorana

**Vary Total System Size**

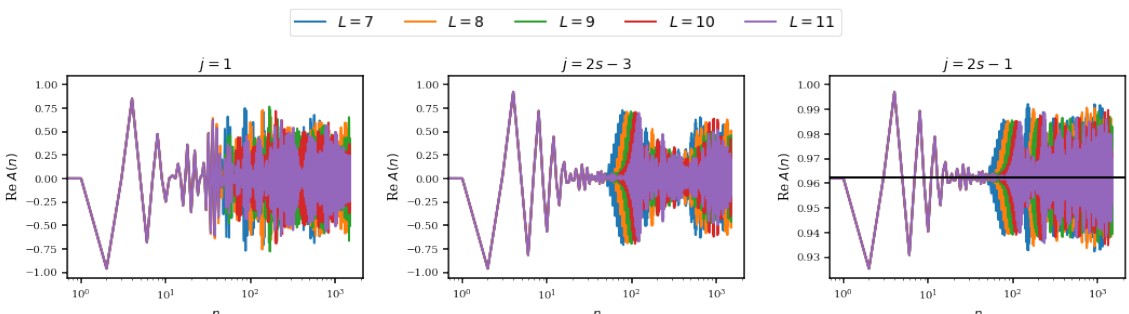

Figure 12: Plots of the real parts of the auto-correlation functions for Majorana fermion $\gamma_{j-1}$ at the odd site $j$ as functions of Floquet periods and for different positions $j = 1, 2s-3, 2s-1$, as the total system size $L$ is varied. The duality-twisted boundary condition is imposed at $2L-1$ and the couplings are set to $u = \frac{\pi}{4}$ and $u' = \frac{\pi}{32}$. The domain wall is located at $2s-1 = 7$ and the qubits live on the odd sites. In the right most plot, the horizontal black line is the amplitude squared of the $\gamma_{2s-2}$ fermion in the Majorana zero mode $\Psi$, in the thermodynamic limit ($L = 100$).

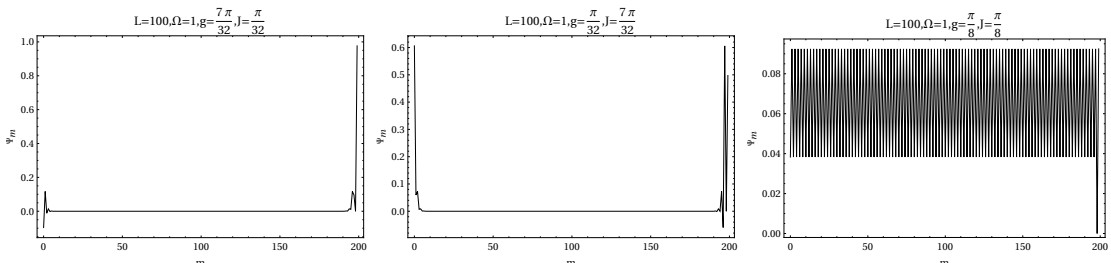

Figure 13: Plots of the zero mode when there is no domain wall. $L = 100$ sites, $\Omega = 1$, and the duality-twisted boundary is located at site $2L-1$. The couplings are chosen so that the left, middle and right plots correspond to the paramagnetic phase, ferromagnetic phase and critical point respectively.

fermions amongst themselves

$$T_\sigma^\dagger \begin{pmatrix} \gamma_{2L-3} \\ \gamma_{2L-1} \\ \gamma_0 \end{pmatrix} T_\sigma = \begin{pmatrix} \cos 2J & -\Omega \sin 2J \cos 2J & \Omega \sin^2 2J \\ \Omega \sin 2J & \cos^2 2J & -\sin 2J \cos 2J \\ 0 & \sin 2J & \cos 2J \end{pmatrix} \begin{pmatrix} \gamma_{2L-3} \\ \gamma_{2L-1} \\ \gamma_0 \end{pmatrix}. \tag{75}$$

The above unitary has an eigenvalue of 1 with a corresponding eigenvector of $(\Omega, \tan J, 1)/\sqrt{1 + \sec^2 J}$.

The zero mode for general values of $g$ and $J$ can be obtained by following the same procedure as in the presence of a domain wall, where we solve the eigenvalue equation $(M - \mathbb{1})\Psi = 0$. This eigenvalue equation leads to the following recursion relation

$$\begin{pmatrix} \Psi_{2j+1} \\ \Psi_{2j+2} \end{pmatrix} = C(g, J)^{j+1} \begin{pmatrix} \Psi_{2L-1} \\ \Psi_0 \end{pmatrix}, \quad j = 0, \ldots, L-3, \tag{76}$$

where the matrix $C(g, J)$ is defined in (55). The last of these equations relate $\Psi_{2L-5}$ and $\Psi_{2L-4}$

to $\Psi_0$ and $\Psi_{2L-1}$. The remaining equations (42c), (42d) and (42e) become

$$0 = \Psi_{2L-4}(\cos 2g \cos 2J - 1) + \Psi_{2L-5} \sin 2J \cos 2g - \Psi_{2L-3} \sin 2g \cos 2J$$
$$+ \Omega \Psi_{2L-1} \sin 2g \sin 2J \cos 2J - \Omega \Psi_0 \sin 2g \sin^2 2J, \tag{77a}$$

$$0 = \Psi_{2L-3}(\cos 2g \cos 2J - 1) + \Psi_{2L-5} \sin 2J \sin 2g + \Psi_{2L-4} \sin 2g \cos 2J$$
$$- \Omega \Psi_{2L-1} \cos 2g \sin 2J \cos 2J + \Omega \Psi_0 \cos 2g \sin^2 2J, \tag{77b}$$

$$0 = \Psi_{2L-1}(\cos^2 2J - 1) - \Psi_0 \sin 2J \cos 2J + \Omega \Psi_{2L-3} \sin 2J. \tag{77c}$$

The final equation allows $\Psi_{2L-3}$ to be written in terms of $\Psi_{2L-1}$ and $\Psi_0$ as

$$\Psi_{2L-3} = (\Psi_{2L-1} \sin 2J + \Psi_0 \cos 2J)\Omega. \tag{78}$$

We will show that the Majorana mode with and without a domain wall share similar asymptotics.

As the eigenvector is only determined up to an overall factor, we can set $\Psi_{2L-1} = 1$ for convenience and normalize the eigenvector at the end. The components $\Psi_{2L-5}$ and $\Psi_{2L-4}$ can be written in terms of $\Psi_0$ and $\Psi_{2L-1} = 1$ by (76), while the component $\Psi_{2L-3}$ can be eliminated using (78). Putting these into (77b) gives

$$\Psi_0 = \frac{\sin 2J \left[ 2\Omega - (\epsilon_{g,J}^{2-L} + \epsilon_{g,J}^{L-2}) \sin 2g \right]}{2\Omega(\cos 2g - \cos 2J) + \sin 2g \left[ \cos 2J(\epsilon_{g,J}^{L-2} + \epsilon_{g,J}^{2-L}) + (\epsilon_{g,J}^{L-2} - \epsilon_{g,J}^{2-L}) \right]}. \tag{79}$$

Above $\epsilon_{g,J}$ is defined in (57). The remaining equation (77a) is satisfied by the above solution, showing that this solution is indeed consistent. Applying the recurrence relation (76) and the relation (78) gives the rest of the coefficients.

Plots of the Majorana zero mode in the absence of a domain wall are shown in figure 13. Away from the critical point, the Majorana zero mode is pinned at the duality-twisted boundary. The main striking difference between this Majorana zero mode and the Majorana zero mode that is pinned on the domain wall is the lack of symmetry for this case. This can already be seen from the analytical solution in the exactly solvable limit $g = 0$, where the symmetry $\Omega$ controls the relative sign between the modes $\Psi_{2L-3}$ and $\Psi_0$ that lie on the left and right of the duality-twisted boundary. As we show below, this asymmetry extends away from the exactly solvable limit.

As a quick check, at the critical point $J = g$, the two solutions for the Majorana zero modes, in the presence and absence of a domain wall, agree. To see this note that when $J = g$, $C(u, u')^n = \mathbb{1}_2$ and

$$\Psi_{2j+1} = 1, \quad j = 0, \dots, L-1; \qquad \Psi_{2j} = \frac{\Omega - \sin 2g}{\cos 2g}, \quad j = 0, \dots, L-2. \tag{80}$$

The above is identical to the solution for the zero mode in the presence of a domain wall as presented in (63), (62), (54b), (56a) and (56b) but with $u = u'$. This agreement is expected as when the couplings are equal, there is no domain wall in the system.

Now we proceed to construct the Majorana zero mode for general couplings. The recurrence relation (76) can be inverted to give

$$\begin{pmatrix} \Psi_{2j-1} \\ \Psi_{2j} \end{pmatrix} = C(g,J)^{-(L-2-j)} \begin{pmatrix} \Psi_{2L-5} \\ \Psi_{2L-4} \end{pmatrix}, \qquad j = 0, \dots, L-3. \tag{81}$$

This is useful for approximating the solution in the thermodynamic limit. First, consider the case where $\frac{\pi}{2} > g > J > 0$ so that $\epsilon_{g,J} > 1$. Therefore, when $L \to \infty$, keeping only the leading order terms in (79) and (78) gives

$$\Psi_0 = -\tan J, \qquad \Psi_{2L-3} = \Omega \tan J, \qquad 0 < J < g < \frac{\pi}{2}. \tag{82}$$

For the modes immediately to the right of the duality-twisted boundary, (76) gives

$$\Psi_{2j+1} = \epsilon_{g,J}^{-j-1}, \qquad \Psi_{2j+2} = -\epsilon_{g,J}^{-j-1}\tan J, \qquad 0 < J < g < \frac{\pi}{2}, j \geq 0. \tag{83}$$

This approximation for $\Psi_{2j+1}$ and $\Psi_{2j+2}$ with $j \geq 0$ does not hold for the modes immediately to the left of the duality-twisted boundary in the thermodynamic limit where $L \to \infty$. To approximate those modes, we have to instead consider the recursion relation that goes in the other direction from the duality-twisted boundary. Substituting (82) into the system of linear equations (77a) and (77b), one finds that $\Psi_{2L-5} = \Omega \tan J \epsilon_{g,J}^{-1}$ and $\Psi_{2L-4} = \Omega \epsilon_{g,J}^{-1}$. Substituting this into (81) gives the approximations to the modes immediately to the left of the duality-twisted boundary

$$\Psi_{2j-1} = \Omega \epsilon_{g,J}^{-(L-1-j)} \tan J, \qquad \Psi_{2j} = \Omega \epsilon_{g,J}^{-(L-1-j)}, \qquad 0 < J < g < \frac{\pi}{2}, j \leq L-2. \tag{84}$$

Next, consider the case where $0 < g < J < \frac{\pi}{2}$ in which case $\epsilon_{g,J} < 1$. Now, the leading order term in (79) and (78) gives

$$\Psi_0 = \cot J, \qquad \Psi_{2L-3} = \Omega \cot J, \qquad 0 < g < J < \frac{\pi}{2}. \tag{85}$$

Applying the recurrence relation (76) gives the exact expression for the modes to the right of the duality-twisted boundary, in the thermodynamic limit

$$\Psi_{2j+1} = \epsilon_{g,J}^{j+1}, \qquad \Psi_{2j+2} = \epsilon_{g,J}^{j+1} \cot J, \qquad 0 < g < J < \frac{\pi}{2}, 0 \leq j. \tag{86}$$

As before, to obtain the thermodynamic limit of the modes to the left of the duality-twisted boundary, we need to instead start from the boundary and move leftwards by applying the inverse recursion relation given by (81). Applying the expressions (85) to the system of linear equations (77a) and (77b) yields $\Psi_{2L-5} = \Omega \epsilon_{g,J} \cot J$ and $\Psi_{2L-4} = -\Omega \epsilon_{g,J}$, and (81) gives the thermodynamic limit of the modes to the left of the duality-twisted boundary

$$\Psi_{2j-1} = \Omega \epsilon_{g,J}^{L-j-1} \cot J, \qquad \Psi_{2j} = -\Omega \epsilon_{g,J}^{L-j-1}, \qquad 0 < g < J < \frac{\pi}{2}, j \leq L-2. \tag{87}$$

In summary, we have shown that in the presence of a duality-twisted boundary, a Majorana zero mode exists. When a domain wall is introduced, the Majorana mode is pinned at the domain wall, decaying away from it symmetrically according to (69) and (72). In the absence of a domain wall, the mode is pinned at the duality-twisted boundary, decaying away from it asymmetrically, and according to (83), (84), (86), (87). Unlike the Majorana zero mode pinned on the domain wall, the choice of different symmetry sectors $\Omega$ can introduce a relative sign between the modes to the left and to the right of the duality-twisted boundary.

Moving away from the domain wall, the coefficients of the Majorana zero mode that is localized on the domain wall get multiplied by either tangent of the smaller coupling or cotangent of the larger coupling. In addition, depending upon which coupling is larger, there is an additional oscillating sign. The Majorana zero mode that is localized at the duality-twisted boundary, in the absence of a domain wall, also depends on the cotangent and the tangent of the couplings, but now it has an oscillating sign on one side of the boundary relative to the other, for all couplings. In particular, for $g > J$, the coefficients to the right of the boundary have oscillating signs, while the coefficients to the left of the boundary do not. This behavior is reversed for $J > g$.

Setting aside the symmetry or asymmetry about the localization position, both zero modes display the same over-all spatial decay away from the localization position. Denoting the uniform couplings by $u_<, u_>$, where $u_< < u_>$, with one of them being the Ising coupling, and the other the transverse-field, the decay length for the Majorana mode with and without the domain wall is identical, and given by $[\ln(\tan u_> / \tan u_<)]^{-1}$. Thus, at the critical point $u_< = u_>$, the Majorana mode is delocalized.

# 6 Conclusions

The study of quantum circuits have so far been restricted to rather simple geometries such as periodic and open boundary conditions. In addition, either the temporal and spatial behavior is ordered, or completely disordered. In this paper we introduced a Floquet circuit which has topological defects, which are new types of inhomogeneities that obey intricate algebraic structures. These defects can be deformed in the spatial and temporal directions without changing the physics, and among them they obey non-trivial fusion rules. We explicitly constructed the circuits with topological defects for the Floquet Ising model. In particular, we presented the Floquet unitaries for a defectless circuit ((2a) and (2b)), a circuit with anti-periodic boundary conditions ((21a) and (21b)), a circuit with duality-twisted boundary conditions ((28a) and (28b)), as well as the spin-flip symmetry operator (8) and the duality defect operator (10). The duality defect is a non-unitary topological defect that performs the Kramers-Wannier duality transformation on the Floquet circuit. We verified explicitly how the fusion algebra of the defects manifest in certain return amplitudes. The twisted boundary conditions amount to arranging the "space-like" spin-flip and duality defect operators in the "time-like" direction. We showed that the duality-twisted boundary conditions allow the system to a host an isolated Majorana zero mode. We analytically constructed the Majorana zero mode and showed how it manifests in the auto-correlation function. When a domain wall is present, this mode is localized at the domain wall, which can also be seen in the continuum limit in the Ising CFT with a space-dependent mass deformation. In the absence of the domain wall, the Majorana mode is localized at the duality-twisted boundary. At the critical point where all couplings are equal, the Majorana mode is delocalized. In contrast to strong zero modes [55], the Majorana mode encountered here is a symmetry of the Floquet circuit.

Future directions involve a classification of Floquet SPTs taking into account topological defects (which are generally described by fusion categories). Understanding the role of interactions is also an important next step. We expect that certain kinds of interactions, while making the circuit non-integrable, will not destroy the topological nature of the defects. For such models, topological defects will be totally immune to heating. It is also interesting to consider interactions for which the defects are no longer exactly topological. For this case it could be that the time scales over which the non-commutativity of the defects are visible, are non-perturbatively long in the interactions. In [91], the effect of integrability breaking interactions which do not preserve the defect commutation relations, was studied. It was found that the isolated Majorana zero mode with duality twisted boundary conditions, was still remarkably robust for small system sizes. This was because for the Majorana mode to decay, the chain has to act as an ideal reservoir, and this requires the chain to be sufficiently long. Thus, finite size effects in fact make the isolated Majorana zero mode more stable. This is in contrast to strong zero modes with open boundary conditions [57,82] where the modes appear in pairs, and are therefore more unstable for smaller system sizes as the modes decay by hybridizing with each other.

Finally, performing braiding in Floquet circuits and exploring the role of junctions on Floquet dynamics, is another important topic to explore in the future. Noisy intermediate scale quantum devices appear to be a promising platform for realizing topological defects. In fact, the duality twisted Floquet unitary was recently simulated on an IBM quantum device [92].

## Acknowledgements

The authors are deeply indebted to Paul Fendley for many helpful discussions and critical comments on the manuscript.

**Funding information** This work was supported by the US National Science Foundation Grant NSF DMR-2018358 (MT, AM). MT is also supported by an appointment to the YST Program at the APCTP through the Science and Technology Promotion Fund and Lottery Fund of the Korean Government, as well as the Korean Local Governments - Gyeongsangbuk-do Province and Pohang City. AM acknowledges the Aspen Center for Physics where part of this work was performed, and which is supported by the National Science Foundation grant PHY-1607611. YW acknowledges support from New York University and the Simons Junior Faculty Fellows program from the Simons Foundation.

## A  Two-qubit Example: Entanglement entropy

In this section, a two-qubit circuit is studied as a warm-up. The small system size makes the expressions for the unitary circuits with the different defects present easier to understand. Furthermore, the entanglement entropy after a single Floquet step can be computed analytically to gain some insight into the effects of the topological defects. In this appendix, the $|\sigma\rangle$ label for the dual sites will be omitted when it is clear to do so.

### A.0.1  Single time step with no defects

We begin with the simplest example with two physical spins and a single time step with no defects. For this case the unitaries are

$$U_{\mathbb{1},\text{even}} = e^{-iu_0 X_0} e^{-iu_2 X_2} e^{-i(u_1+u_3)Z_0 Z_2}, \tag{88a}$$

$$U_{\mathbb{1},\text{odd}} = e^{-i(u_0+u_2)Z_1 Z_3} e^{-iu_1 X_1} e^{-iu_3 X_3}. \tag{88b}$$

The Floquet unitaries have different orderings of the transverse field and Ising coupling unitaries depending on whether the physical spins are on even or odd lattices. For an initial product state in the $z$ basis $|\Psi(0)\rangle = |h_0 h_2\rangle$ for the even lattice and $|\Psi(0)\rangle = |h_1 h_3\rangle$ for the odd lattice, the entanglement entropy of a single site after a single step of the defectless circuit is

$$S_{\{0\}} = 0, \qquad S_{\{1\}} = -\lambda_+ \ln \lambda_+ - \lambda_- \ln \lambda_-. \tag{89}$$

Above $S_{\{0\}}$ and $S_{\{1\}}$ are the single site entanglement entropies for the even (site 0) and odd (site 1) lattices respectively and

$$\lambda_\pm = \frac{1}{2} \left[ 1 \pm \sqrt{1 - \sin^2 2u_1 \sin^2 2u_3 \sin^2 2(u_0+u_2)} \right], \tag{90}$$

are the eigenvalues of the reduced density matrix for site 1. The single site entanglement entropy vanishes for spins on the even lattice, as long as the initial state is a product state in the $z$ basis. This is because applying the Ising coupling term to such a product state merely produces a phase, while the subsequent on-site magnetic fields rotate the two spins independently so no entanglement entropy is generated. For spins on the odd lattice, the onsite magnetic fields are applied first, rotating the spins that were initially product states in the $z$ basis. Applying the Ising coupling gates after this rotation then generates entanglement entropy.

### A.0.2  Single time step with Anti-periodic boundary conditions

Consider a two-qubit circuit with anti-periodic boundary conditions imposed between sites 1 and 2, i.e. $s = 1$ in (20). The Floquet unitaries are given by

$$T_{\psi,1,\text{even}} = e^{-iu_0 X_0} e^{-iu_2 X_2} e^{i(u_1-u_3)Z_0 Z_2}, \tag{91a}$$

$$T_{\psi,1,\text{odd}} = e^{i(u_2-u_0)Z_1 Z_3} e^{-iu_1 X_1} e^{-iu_3 X_3}. \tag{91b}$$

Comparing this with the defectless unitary (88), we see that the effect of the anti-periodic boundary condition is to flip the signs of the Ising couplings in $e^{-iu_1 Z_0 Z_2}$ and $e^{-iu_2 Z_1 Z_3}$ for the even and odd lattice respectively.

Let us consider evolving initial states that are product states in the $z$ basis, $|\Psi(0)\rangle = |h_0 h_2\rangle$ for the even lattice and $|\Psi(0)\rangle = |h_1 h_3\rangle$ for the odd lattice. After a single step of the unitary circuit with anti-periodic boundary conditions, the single site entanglement entropy is

$$S_{\{0\}} = 0, \qquad S_{\{1\}} = -\lambda_+ \log \lambda_+ - \lambda_- \log \lambda_-, \tag{92}$$

for the even and odd lattices respectively. Moreover

$$\lambda_\pm = \frac{1}{2} \left[ 1 \pm \sqrt{1 - \sin^2 2u_1 \sin^2 2u_3 \sin^2 2(u_2 - u_0)} \right], \tag{93}$$

are the eigenvalues of the reduced density matrix for site 1 of the odd lattice. The single site entanglement entropy is zero for the even lattice for the very same reason as the defectless circuit because the anti-periodic boundary condition flips the signs of one of the Ising couplings, and hence only affects the initial phase engendered by the Ising coupling unitaries.

The single-site entanglement entropy for the odd lattice does not depend on the initial state when it is a product state in the $z$ basis as it only affects certain phases in the state. The entanglement entropy is the same as that for the defectless unitary except that one of the Ising couplings has its sign flipped. As a result, if $u_2 = u_0$, the entanglement entropy vanishes. This is because the anti-periodic boundary condition flipped the sign on one of the Ising unitaries and so if $u_2 = u_0$, the phases coming from the two Ising unitaries cancel out and the state remains a product state.

### A.0.3 Single time step with Duality defect

Consider a single Floquet period with a duality defect sandwiched between the two steps of the Floquet drive. The matrix element when the input spins are placed on the even sites is

$$\langle h'_1 h'_3 | U_{\text{even}} | h_0 h_2 \rangle = \quad \text{} \tag{94}$$

The matrix element when the input spins are on the odd sites is similar. The unitary operators for a single time step acting on $\mathcal{H}_{\text{even}}$ and $\mathcal{H}_{\text{odd}}$ are

$$U_{\text{even}} = e^{-iu_0 Z_1 Z_3} e^{-iu_2 Z_1 Z_3} D_\sigma e^{-iu_1 Z_0 Z_2} e^{-iu_3 Z_0 Z_2}, \tag{95a}$$

$$U_{\text{odd}} = e^{-iu_0 X_0} e^{-iu_2 X_2} D_\sigma e^{-iu_1 X_1} e^{-iu_3 X_3}. \tag{95b}$$

The duality defect exchanges each lattice with its dual. If the input state lives on the even (odd) lattice, the output state will live on the odd (even) lattice. By comparing this with the defectless unitary (88), we see that the net effect of this transformation is to turn Ising coupling unitaries into transverse field unitaries and vice versa. To gain better intuition about the action of the duality defect, consider an initial state $|00\rangle$ on the even sites and with all couplings set

to $u_i = u$. Under this Floquet unitary, the state evolves as

$$|00\rangle \xrightarrow{\text{Ising Coupling}} e^{-2iu}|00\rangle \tag{96a}$$

$$\xrightarrow{\text{Duality Defect}} e^{-2iu}|++\rangle = e^{-2iu}\frac{|00\rangle + |01\rangle + |10\rangle + |11\rangle}{2} \tag{96b}$$

$$\xrightarrow{\text{Ising Coupling}} e^{-4iu}\frac{|00\rangle + |11\rangle}{2} + \frac{|01\rangle + |10\rangle}{2}. \tag{96c}$$

The initial Ising coupling term generated no entanglement since $|00\rangle$ remained a product state. After the duality defect, the system is still in a product state but in the $x$ basis instead. Now, acting on the system with the Ising coupling terms will generate entanglement. On the other hand, if the initial state $|00\rangle$ is defined on the odd lattice, the state will evolve under a single application of the Floquet unitary as

$$|00\rangle \xrightarrow{\text{on-site magnetic fields}} \cos^2 u|00\rangle - \sin^2 u|11\rangle - i\sin u\cos u(|01\rangle + |10\rangle)) \tag{97a}$$

$$\xrightarrow{\text{Duality Defect}} \cos 2u|++\rangle - i\sin 2u|--\rangle \tag{97b}$$

$$\xrightarrow{\text{on-site magnetic fields}} \cos 2ue^{-i2u}|++\rangle - i\sin 2ue^{i2u}|--\rangle. \tag{97c}$$

The first row of on-site magnetic fields rotates the two spins. The duality defect maps aligned (anti-aligned) spins to spins that are polarized (anti-polarized) along the magnetic field. After the action of the duality defect, the state is no longer separable because the duality defect has implemented a projection.

Next, consider more general initial product states $|\Psi(0)\rangle = |h_0 h_2\rangle$ and $|\Psi(0)\rangle = |h_1 h_3\rangle$. The single-site entanglement entropy is

$$
\begin{aligned}
&S_{\{1\}}\\
&= -\frac{1}{2}\log\frac{1 - \cos^2[2(u_0 + u_2)]}{4} - \frac{(-1)^{h_0+h_2}}{2}\cos[2(u_0 + u_2)]\log\frac{1 + (-1)^{h_0+h_2}\cos[2(u_0+u_2)]}{1 - (-1)^{h_0+h_2}\cos[2(u_0+u_2)]},
\end{aligned}
\tag{98a}
$$

$$
\begin{aligned}
&S_{\{0\}}\\
&= -\frac{1}{2}\log\frac{1 - \cos^2[2(u_1 + u_3)]}{4} - \frac{(-1)^{h_1+h_3}}{2}\cos[2(u_1 + u_3)]\log\frac{1 + (-1)^{h_1+h_3}\cos[2(u_1+u_3)]}{1 - (-1)^{h_1+h_3}\cos[2(u_1+u_3)]},
\end{aligned}
\tag{98b}
$$

for the initial states living on the even and odd lattices respectively. Note that the entanglement entropy only depends on the Ising couplings $u_0, u_2$ when the input sites live on the even lattice, and on $u_1, u_3$ when the input sites live on the odd lattice. The easiest way to understand this is to use the duality defect commutation relations (12) to drag the duality defect all the way to the bottom. This will transform the initial state from a product state in the $z$ basis to a product state in the $x$ basis. Also, the transverse magnetic fields for the second case with the input state living on the odd sites turn into Ising coupling unitaries. Then, the only couplings that appear in the single-site entanglement entropy for this transformed initial state (product state in the $x$ basis) are the Ising couplings since the transverse magnetic field merely rotates individual spins in this basis.

### A.0.4 Single time step with spin-flip defect

Next, consider inserting a spin-flip defect in between the two-steps of the Floquet unitary. The matrix element for the unitary that acts on $\mathcal{H}_{\text{even}}$ is

$$\langle h'_0 h'_2 | U_{\text{even}} | h_0 h_2 \rangle = \quad \text{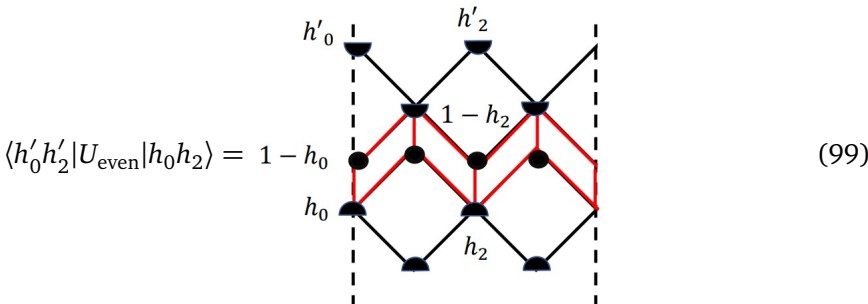} \tag{99}$$

The diagram depicting the matrix element for the unitary acting on $\mathcal{H}_{\text{odd}}$ is similar. The unitary operator is found to be

$$U_{\text{even}} = e^{-iu_0 X_0} e^{-iu_2 X_2} X_0 X_2 e^{-i(u_1+u_3)Z_0 Z_2}, \tag{100a}$$

$$U_{\text{odd}} = e^{-i(u_0+u_2)Z_1 Z_3} X_1 X_3 e^{-iu_1 X_1} e^{-iu_3 X_3}. \tag{100b}$$

The unitary operator is identical to the defectless case (88) except for the additional spin-flip defect which is the Ising symmetry operator. The Ising symmetry operator can be dragged to the bottom of the circuit by applying the defect commutation relation (12). Action of the spin-flip operator on an initial product state in the $z$ basis simply flips each spin. Since the single site entanglement entropy for these initial states do not depend on the particular $z$ basis state, it will be identical to that of the defectless circuit (89). This is to be expected since the spin-flip defect is the Ising symmetry of the model and should not affect physical quantities like the entanglement entropy.

### A.0.5 Single time step with duality-twisted boundary conditions

Finally, consider a single time step with duality-twisted boundary conditions. The corresponding unitary operator for a single time-step is represented by the diagram

$$\langle h'_0\sigma h'_2\sigma|T_{\sigma,\text{even}}|h_0\sigma h_2\sigma\rangle = \sum_\beta \qquad\qquad\qquad\qquad\qquad\qquad\qquad (101\text{a})$$

$$\langle \sigma h'_1\sigma h'_3|T_{\sigma,\text{odd}}|\sigma h_1\sigma h_3\rangle = \qquad\qquad\qquad\qquad\qquad\qquad\qquad (101\text{b})$$

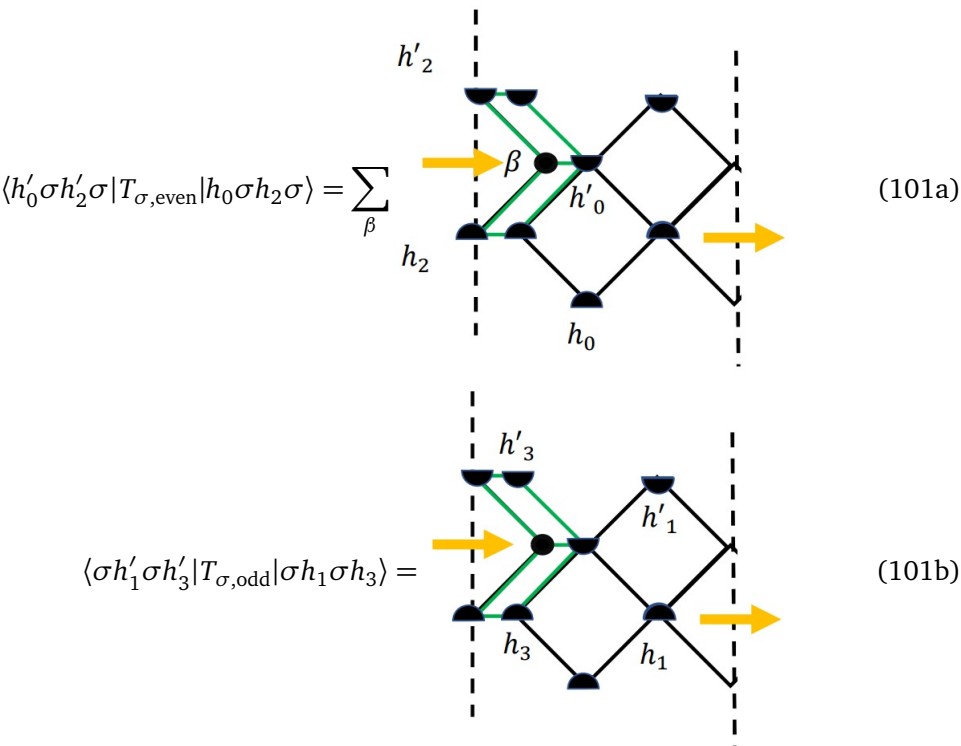

Applying the matrix elements gives the unitary operator

$$T_{\sigma,\text{even}} = e^{-iu_1 Z_0 Z_2} CZ_{0,2} e^{-iu_2 X_2} CZ_{0,2} e^{-iu_0 X_0}, \qquad (102\text{a})$$

$$T_{\sigma,\text{odd}} = e^{-iu_1 X_1} H_3 e^{-iu_2 Z_1 Z_3} H_3 e^{-iu_0 Z_1 Z_3}. \qquad (102\text{b})$$

Conjugating an Ising coupling unitary and a transverse field unitary with Hadamard gates and controlled Z gates leads to mixed coupling terms. Thus, the Floquet unitary with duality-twisted boundary conditions takes the form

$$T_{\sigma,\text{even}} = e^{-iu_1 Z_0 Z_2} e^{-iu_2 Z_0 X_2} e^{-iu_0 X_0}, \qquad (103\text{a})$$

$$T_{\sigma,\text{odd}} = e^{-iu_1 X_1} e^{-iu_2 Z_1 X_3} e^{-iu_0 Z_1 Z_3}. \qquad (103\text{b})$$

To better understand the effect of such a circuit, consider its action on a simple initial product state $|00\rangle$. When this state is placed on the odd sites, the homogeneous choice of couplings $u_i = \frac{\pi}{4}$ results in

$$|00\rangle \xrightarrow{\text{Ising Coupling}} |00\rangle \xrightarrow{\text{Mixed Coupling}} |0\rangle\frac{|0\rangle - i|1\rangle}{\sqrt{2}}$$
$$\xrightarrow{\text{Transverse Field}} \frac{|0\rangle - i|1\rangle}{\sqrt{2}}\frac{|0\rangle - i|1\rangle}{\sqrt{2}}. \qquad (104)$$

This is a product state so the entanglement entropy of either site is 0. On the other hand, when the physical spins are placed on the even sites, the initial product state transforms as

$$|00\rangle \xrightarrow{\text{Transverse Field}} \frac{|0\rangle - i|1\rangle}{\sqrt{2}}|0\rangle \xrightarrow{\text{Mixed coupling}} \frac{|00\rangle - i|01\rangle - i|10\rangle + |11\rangle}{2}$$
$$\xrightarrow{\text{Ising Coupling}} \frac{e^{-i\frac{\pi}{4}}(|00\rangle + |11\rangle) - ie^{i\frac{\pi}{4}}(|01\rangle + |10\rangle)}{2}. \qquad (105)$$

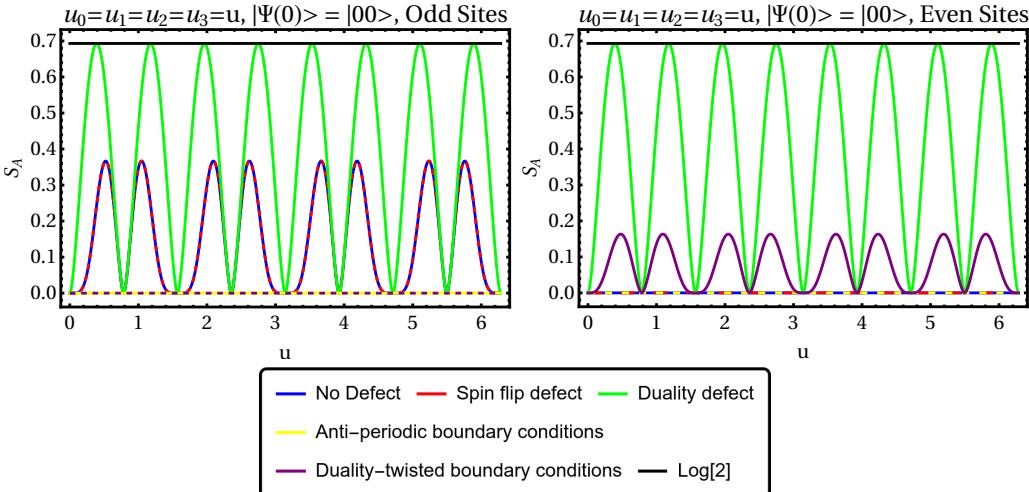

Figure 14: Plots of the half-chain entanglement entropy after a single time step against the homogeneous coupling $u_i = u$, and for different defects present. We take a chain of length two with periodic boundary conditions and for an initial state that is a product state $|h_i h_j\rangle$ in the $z$ basis. The entanglement entropy does not depend on $h_i, h_j$. Left Plot: The physical spins live on the odd sites. The entanglement entropy for the circuit with anti-periodic and duality-twisted boundary conditions is zero, while the entanglement entropy for the circuit with a spin-flip defect is identical to that of the defectless circuit. Right Plot: The physical spins live on the even sites. The entanglement entropy for a circuit with a spin-flip defect or with anti-periodic boundary conditions vanishes just like for the defectless circuit.

More generally, for an arbitrary $z$ basis product state $|\Psi(0)\rangle = |h_i h_{i+2}\rangle$, the resulting state after applying a single step of the Floquet unitary with a duality-twisted boundary is

$$T_{\sigma,\text{odd}}|h_1 h_3\rangle = e^{-iu_0(-1)^{h_1+h_3}} \left( \cos u_1 |h_1\rangle - i\sin u_1 |1-h_1\rangle \right)\left( \cos u_2 |h_3\rangle - i\sin u_2 (-1)^{h_1}|1-h_3\rangle \right), \tag{106}$$

$$\begin{aligned} T_{\sigma,\text{even}}|h_0 h_2\rangle =& \cos u_0 \cos u_2 e^{-iu_1(-1)^{h_0+h_2}}|h_0 h_2\rangle - i\sin u_0 \cos u_2 e^{iu_1(-1)^{h_0+h_2}}|1-h_0, h_2\rangle \\ &-i\cos u_0 \sin u_2 (-1)^{h_0} e^{iu_1(-1)^{h_0+h_2}}|h_0, 1-h_2\rangle \\ &+\sin u_0 \sin u_2 (-1)^{h_0} e^{-iu_1(-1)^{h_0+h_2}}|1-h_0, 1-h_2\rangle. \end{aligned} \tag{107}$$

The single-site entanglement entropy after a single application of this unitary operator with inhomogeneous couplings is

$$S_{\{1\}} = 0, \qquad S_{\{0\}} = -\lambda_+ \ln \lambda_+ - \lambda_- \ln \lambda_-, \tag{108}$$

for the odd and even lattices respectively. Moreover,

$$\lambda_\pm = \frac{1}{2}\left[ 1 \pm \sqrt{1 - \cos^2 2u_1 \sin^2 2u_0 \sin^2 2u_2} \right], \tag{109}$$

are the eigenvalues of the reduced density matrix on site 0.

### A.0.6 Summary of two-site entanglement entropy after a single time step

To better understand the effects of the topological defects, the half-chain entanglement entropy for a system with two physical qubits, and after a single time-step of a Floquet unitary with

different defect insertions, is shown in figure 14. For concreteness, the initial state is taken to be a product state in the $z$ basis. In this basis, entanglement entropy can be generated when an Ising coupling term is applied after transverse magnetic fields as the latter will rotate the various spins and the former will produce different phases for different states in the $z$ basis.

The simplest defect is the spin-flip defect as it is the Ising symmetry generator which flips the initial product state, and has no bearing on the single site entanglement entropy. The anti-periodic boundary conditions on the other hand reverses the sign of some of the Ising couplings. The resulting phases that are generated by neighbouring Ising couplings cancel each other out, leading to a reduction in the final single-site entanglement entropy.

The most interesting topological defect is the duality defect as it is a symmetry operator of the system that is non-unitary. Its effect is to change the initial state which is polarized in the $z$ direction to one that is a product state in the $x$ basis. This can produce entanglement because operators that do not generate entanglement in one basis can do so in another basis.

## B  Proof of (64)

By inverting (54a), the Majorana modes to the left of $\Psi_{2s-2}$ can be written as

$$
\begin{pmatrix} \Psi_{2j-1} \\ \Psi_{2j} \end{pmatrix} = C(u,u')^{-(s-1-j)} \begin{pmatrix} \Psi_{2s-3} \\ \Psi_{2s-2} \end{pmatrix}, \qquad j = 0, \ldots, s-2, \tag{110}
$$

with the powers of the inverse matrix given by

$$
C(u,u')^{-n} = \begin{pmatrix} \epsilon_{u,u'}^{n} \cos^2 u' + \epsilon_{u,u'}^{-n} \sin^2 u' & (\epsilon_{u,u'}^{-n} - \epsilon_{u,u'}^{n}) \sin u' \cos u' \\ (\epsilon_{u,u'}^{-n} - \epsilon_{u,u'}^{n}) \sin u' \cos u' & \epsilon_{u,u'}^{n} \sin^2 u' + \epsilon_{u,u'}^{-n} \cos^2 u' \end{pmatrix}, \tag{111}
$$

for positive integers $n$. The first row of (54b) implies $\Psi_{2s-3} = \Psi_{2s-1}$, and together with the second row of (54b) one can write $\Psi_{2s}$ in terms of $\Psi_{2s-1}$ and $\Psi_{2s-2}$ as follows

$$
\Psi_{2s} = \left( \cot 2u - \frac{\cos 2u'}{\sin 2u} \right) \Psi_{2s-1} + \frac{\sin 2u'}{\sin 2u} \Psi_{2s-2}. \tag{112}
$$

Substituting this into (56b), the recursion relations for the Majornana fermions on the right of the domain wall, and using the double angle formula $\cos 2u - \cos 2u' = 2\cos^2 u - 2\cos^2 u' = 2\sin^2 u' - 2\sin^2 u$, one finds

$$
\Psi_{2j+1} = (\epsilon_{u,u'}^{s-j-1} \sin^2 u' + \epsilon_{u,u'}^{j-s+1} \cos^2 u') \Psi_{2s-1} + (\epsilon_{u,u'}^{s-j-1} - \epsilon_{u,u'}^{j-s+1}) \sin u' \cos u' \Psi_{2s-2}, \tag{113a}
$$

$$
\Psi_{2j+2} = (\epsilon_{u,u'}^{s-2-j} - \epsilon_{u,u'}^{j-s+2}) \sin u' \cos u' \Psi_{2s-1} + (\epsilon_{u,u'}^{j-s+2} \sin^2 u' + \epsilon_{u,u'}^{s-2-j} \cos^2 u') \Psi_{2s-2}, \tag{113b}
$$

for $j = s, \ldots, L-3$. By shifting the index $j$, the second equation, (113b), can be re-written as

$$
\Psi_{2j} = (\epsilon_{u,u'}^{s-1-j} - \epsilon_{u,u'}^{j-s+1}) \sin u' \cos u' \Psi_{2s-1} + (\epsilon_{u,u'}^{j-s+1} \sin^2 u' + \epsilon_{u,u'}^{s-1-j} \cos^2 u') \Psi_{2s-2}, \tag{114}
$$

for $j = s+1, \ldots, L-2$. When $j = s$, this equation agrees with (112) so it holds for $j = s$ as well. Therefore, (113a) and (114) can be packaged into a single matrix equation

$$
\begin{pmatrix} \Psi_{2j+1} \\ \Psi_{2j} \end{pmatrix} = C(u,u')^{-(j-s+1)} \begin{pmatrix} \Psi_{2s-1} \\ \Psi_{2s-2} \end{pmatrix}, \qquad j = s, \ldots, L-3. \tag{115}
$$

Since $\Psi_{2s-3} = \Psi_{2s-1}$, the above equation is essentially identical to (110) except that it applies to the Majorana fermions to the right of the domain wall. Thus we have explicitly shown that the zero mode solution is symmetric about the domain wall.

# C   Topological defects in the Ising CFT and Majorana modes

Conventional global symmetries are associated with unitary operators that commute with the Hamiltonian. These unitary operators act linearly on the Hilbert space, implementing the symmetry transformations. Recently it was realized that there is a natural generalized notion of symmetries by topological defect operators defined on closed oriented submanifolds of spacetime [93] (see also [68, 69] for a recent review), where the conservation of the symmetry charge is encoded in the topological property of the defect operator, and the composition of symmetry transformations is represented by the fusion product of topological defects. The most general topological defects define the generalized symmetries. In this language, the conventional global symmetries constitute the special case where the topological defects are extended along the entire space and invertible, meaning that their fusion products follow group multiplication and a symmetry transformation can be undone by inserting a defect representing its inverse. For general topological defects, an inverse does not exist, in which case we call the defect non-invertible and the corresponding generalized symmetry is a non-invertible symmetry. Equivalently their fusion products no longer form a group, but instead a fusion algebra, where a direct sum of topological defects appear from the fusion of a pair.

In the following we will a give brief review of the topological defects (equivalently generalized symmetries) and their underlying structures in the Ising CFT, as well as their descriptions in the related theory of a Majorana fermion. We then use these structures to discuss the continuum version of the Majorana modes found in Section 5.1.

## C.1   Review of Ising topological defects

The Ising CFT has three topological defects

$$D_{\mathbb{1}}, \quad D_{\psi}, \quad D_{\sigma}, \tag{116}$$

that obey the following fusion rules

$$D_{\psi}D_{\psi} = D_{\mathbb{1}}, \quad D_{\sigma}D_{\sigma} = D_{\mathbb{1}} + D_{\psi}, \quad D_{\sigma}D_{\psi} = D_{\psi}D_{\sigma} = D_{\sigma}. \tag{117}$$

In particular $D_{\mathbb{1}}$ is the transparent (trivial) line defect and $D_{\psi}$ is the invertible defect that generates the $\mathbb{Z}_2^C$ spin-flip (charge-conjugation) symmetry in the Ising model.

In general, topological defects may form $n$-fold topological junctions (with $n$ incoming topological defect legs). A general $n$-fold topological junction can be resolved into multiple trivalent junctions. Different resolutions lead to different basis for the vector space of $n$-fold topological junctions.[3] For example, for a four-fold junction, there are two resolutions related by a F-move and the change of basis matrix is given by the F-symbol $(F_d^{abc})_{xy}$ where $a, b, c, d$ label the four external defects and $x, y$ label the internal defects that appear in the two resolution channels respectively (see figure 15). The F-symbols are subjected to constraints from consistency conditions of the resolutions of $n$-fold junctions (with $n \geq 5$). In particular the famous pentagon relations arise from $n = 5$-fold junctions, which turn out to be necessary and sufficient to ensure the consistency for all topological junctions. Each solution to the pentagon relations (up to a certain "gauge freedom") for the set of topological defects we start with fully specifies a fusion category, which is the mathematical framework underlying generalized symmetries in $1 + 1d$ (see [94] for more about fusion categories).

For the topological defects in (116) obeying the fusion rules (117), there are two solutions to the pentagon equations that differ by an overall sign in the F-symbol $F_{\sigma}^{\sigma\sigma}$. One of them

---

[3]In CFT, this topological junction vector space is the subspace of the defect Hilbert space (on a $S^1$ with $n$ defect points) of zero conformal dimension.

describes the generalized symmetry in the Ising CFT, and for that reason, we call the corresponding fusion category the Ising category. The other describes a generalized symmetry in the $SU(2)_2$ WZW model. We record the nontrivial F-symbols for the Ising category below

$$(F_\sigma^{\sigma\sigma\sigma})_{xy} = \frac{1}{\sqrt{2}} \begin{pmatrix} 1 & 1 \\ 1 & -1 \end{pmatrix}, \quad F_\psi^{\sigma\psi\sigma} = -1\,. \tag{118}$$

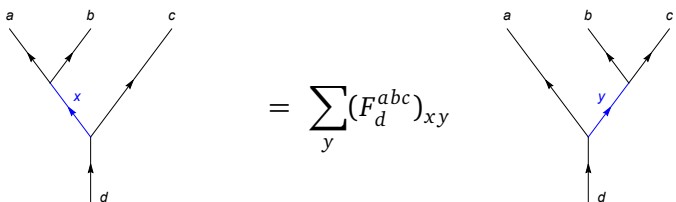

Figure 15: The F-move that relates two resolutions of the four-fold junction of topological defects labelled by $a, b, c, d$ into a pair of three-fold junctions joined by internal topological defects (in blue) labelled by $x$ and $y$. Here the arrows keep track of the orientations of the defects.

In CFT, topological defects have richer structures encoding how they act on local operators (equivalently states on $S^1$). Firstly, a topological defect $\mathcal{L}$ when positioned along a spatial direction naturally act on the Hilbert space $\mathcal{H}$ of the CFT on $S^1$ as a linear operator $\widehat{\mathcal{L}}$ (which is not unitary in general). Secondly, a topological defect $\mathcal{L}$ when positioned along the time direction defines a *twisted* Hilbert space $\mathcal{H}_\mathcal{L}$ (e.g. $\mathcal{H}_\mathbb{1} = \mathcal{H}$) also known as the defect Hilbert space for $\mathcal{L}$. More generally, topological defects $\mathcal{L}_i$ also act on defect Hilbert spaces $\mathcal{H}_{\mathcal{L}_j}$. In this case, since the defects $\mathcal{L}_i$ and $\mathcal{L}_j$ intersect on the cylinder, the corresponding linear operator acting on $\mathcal{H}_{\mathcal{L}_j}$ in general depend on the junction vector (equivalently a choice of resolution into a pair of trivalent junctions). Finally, since topological defects commute with the Virasoro generators, to specify these linear operators acting on various Hilbert spaces on $S^1$, it suffices to state their actions on the Virasoro primaries in each sector.

We now review these defect actions in the Ising CFT (see [95] for general discussions on topological defects in $1+1d$ CFTs). In the Ising CFT, the (twisted) Hilbert space organize into unitary representations of the Virasoro algebras $\text{Vir}_{c=1/2} \times \overline{\text{Vir}}_{c=1/2}$ which may have (anti)chiral weights $h, \bar{h} \in \{0, \frac{1}{2}, \frac{1}{16}\}$. In the ordinary Hilbert space $\mathcal{H}$, the primaries are the identity operator $\mathbb{1}$, the energy operator $\epsilon$ of dimension $h = \bar{h} = \frac{1}{2}$, and the spin operator $\sigma$ with $h = \bar{h} = \frac{1}{16}$. Below we list the primaries in general defect Hilbert spaces where the subscripts keep track of the $(h, \bar{h})$ weights of the corresponding operators,

$$\mathcal{H} = \{\mathbb{1}_{0,0}, \epsilon_{\frac{1}{2},\frac{1}{2}}, \sigma_{\frac{1}{16},\frac{1}{16}}\}, \quad \mathcal{H}_\psi = \{\psi_{\frac{1}{2},0}, \bar{\psi}_{0,\frac{1}{2}}, \mu_{\frac{1}{16},\frac{1}{16}}\}, \quad \mathcal{H}_\sigma = \{s_{\frac{1}{16},0}, \bar{s}_{0,\frac{1}{16}}, \Lambda_{\frac{1}{16},\frac{1}{2}}, \bar{\Lambda}_{\frac{1}{2},\frac{1}{16}}\}\,. \tag{119}$$

In particular, in the $\mathbb{Z}_2^C$ twisted Hilbert space $\mathcal{H}_\psi$, $\mu$ is the disorder spin operator and $\psi, \bar{\psi}$ are the left and right moving Majorana fermions. Perhaps less known are the operators in the $D_\sigma$ twisted Hilbert space $\mathcal{H}_\sigma$ listed in the last equality above. They (together with the Virasoro descendants) represent a basis of operators that live at the end of the duality defect $D_\sigma$. The topological defects act on a (twisted) point-like operator by enclosing it as in Fig. 16. Equivalently this determines how they act on the states in the (twisted) Hilbert spaces. More generally, topological defects relate states in Hilbert spaces (correspondingly point-like operators) with different twists. See Fig. 17 for an example in the Ising CFT which is an important feature of the duality defect $D_\sigma$.

The action of the topological defects on (twisted) Hilbert spaces in the Ising CFT is summarized in Table 1. Note that the linear operators $\widehat{D}_a$ in general do not obey the same fusion

rule as in (117) when acting on a twisted Hilbert space (this is a consequence of the nontrivial F-symbols).

| $\mathcal{H}_{\mathbb{1}}$ | $\mathbb{1}$ | $\epsilon$ | $\sigma$ |
|---|---|---|---|
| $\widehat{D}_{\mathbb{1}}$ | 1 | 1 | 1 |
| $\widehat{D}_{\psi}$ | 1 | 1 | $-1$ |
| $\widehat{D}_{\sigma}$ | $\sqrt{2}$ | $-\sqrt{2}$ | 0 |

| $\mathcal{H}_{\psi}$ | $\psi$ | $\bar{\psi}$ | $\mu$ |
|---|---|---|---|
| $\widehat{D}_{\mathbb{1}}$ | 1 | 1 | 1 |
| $\widehat{D}_{\psi}$ | $-1$ | $-1$ | 1 |
| $\widehat{D}_{\sigma}$ | $-\sqrt{2}i$ | $\sqrt{2}i$ | 0 |

| $\mathcal{H}_{\sigma}$ | $s$ | $\bar{s}$ | $\Lambda$ | $\bar{\Lambda}$ |
|---|---|---|---|---|
| $\widehat{D}_{\mathbb{1}}$ | 1 | 1 | 1 | 1 |
| $\widehat{D}_{\psi}$ | $i$ | $-i$ | $i$ | $-i$ |
| $\widehat{D}_{\sigma}$ | $e^{\frac{\pi i}{8}}$ | $e^{-\frac{\pi i}{8}}$ | $-e^{\frac{\pi i}{8}}$ | $-e^{-\frac{\pi i}{8}}$ |

Table 1: The action of topological defects restricted to (twisted) Hilbert spaces on $S^1$ in the Ising CFT. Here the defects act diagonally and the diagonal entries are listed above. We have picked a particular resolution of the nontrivial four-fold topological junctions when the defects act on twisted Hilbert spaces.

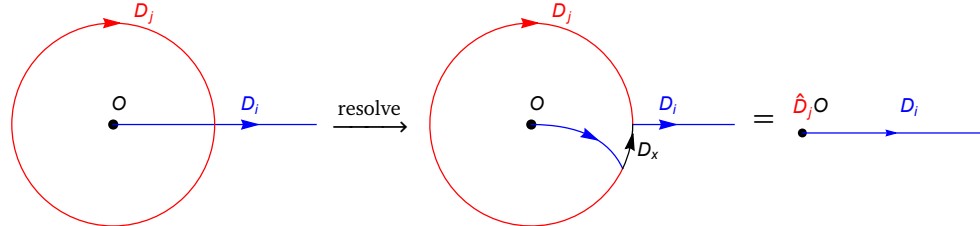

Figure 16: The action of the topological defect $D_j$ on an operator $O$ that lives at the end of the defect $D_i$ (equivalently a state in the defect Hilbert space $\mathcal{H}_i$). The linear map $\widehat{D}_j$ depends implicitly on a choice of the four-fold topological junction (in the first picture above from the left), which is specified by a resolution into three-fold topological junctions via a topological defect $D_x$ (second picture).

Figure 17: As we move the duality defect $D_{\sigma}$ across the order spin operator $\sigma(x)$, the disorder spin operator $\mu(x)$ emerges at the end of the $\mathbb{Z}_2$ spin-flip defect $D_{\psi}$.

The operator content of the Ising CFT in the absence of defects is captured by its torus partition function

$$Z(\tau, \bar{\tau}) = |\chi_{\mathbb{1}}(\tau)|^2 + |\chi_{\psi}(\tau)|^2 + |\chi_{\sigma}(\tau)|^2. \tag{120}$$

Similarly, the defect Hilbert space and the action of the topological defects therein in a general CFT of central charge $c$ follows from the (twisted) torus partition functions $Z_{i|j}(\tau, \bar{\tau})$ decorated by defects $D_i$ and $D_j$ along the time and spatial cycles of the torus respectively (see figure 18),

$$Z_{i|j}(\tau, \bar{\tau}) \equiv \operatorname{tr}_{\mathcal{H}_i}\left(\widehat{D}_j e^{2\pi i \tau (L_0 - \frac{c}{48})} e^{-2\pi i \bar{\tau}(\bar{L}_0 - \frac{c}{48})}\right). \tag{121}$$

The twisted partition functions are constrained by modular covariance, namely under the T-transformation (Dehn twist) and the S-transformation,

$$Z_{i|j}(\tau,\tau) \xrightarrow{T} Z_{i|j}(\tau+1,\bar{\tau}+1) = Z_{i|j\cdot i}(\tau,\bar{\tau}),$$

$$Z_{i|j}(\tau,\tau) \xrightarrow{S} Z_{i|j}(-1/\tau,-1/\bar{\tau}) = Z_{j|\bar{i}}(\tau,\bar{\tau}),$$

(122)

where $\bar{i}$ denotes the orientation-reversal of $D_i$ and $j\cdot k$ denotes the fusion product of $D_j$ and $D_k$.

For the Ising CFT, we record the twisted partition functions below (note that the Ising topological defects are all identical to the orientation reversal of themselves),

$$Z_{\mathbb{1}|\psi} = |\chi_{\mathbb{1}}(\tau)|^2 + |\chi_\psi(\tau)|^2 - |\chi_\sigma(\tau)|^2, \quad Z_{\mathbb{1}|\sigma} = \sqrt{2}|\chi_{\mathbb{1}}(\tau)|^2 - \sqrt{2}|\chi_\psi(\tau)|^2,$$

$$Z_{\psi|\mathbb{1}} = (\chi_{\mathbb{1}}(\tau)\chi_\psi(\bar{\tau}) + \text{c.c.}) + |\chi_\sigma(\tau)|^2, \quad Z_{\sigma|\mathbb{1}} = \chi_\sigma(\tau)(\chi_{\mathbb{1}}(\bar{\tau}) + \chi_\psi(\bar{\tau})) + \text{c.c.},$$

$$Z_{\psi|\psi} = -(\chi_{\mathbb{1}}(\tau)\chi_\psi(\bar{\tau}) + \text{c.c.}) + |\chi_\sigma(\tau)|^2, \quad Z_{\sigma|\psi} = i\chi_\sigma(\tau)(\chi_{\mathbb{1}}(\bar{\tau}) + \chi_\psi(\bar{\tau})) + \text{c.c.},$$

$$Z_{\psi|\sigma} = -i\sqrt{2}\chi_\psi(\tau)\chi_{\mathbb{1}}(\bar{\tau}) + \text{c.c.}, \quad Z_{\sigma|\sigma} = e^{\frac{\pi i}{8}}\chi_\sigma(\tau)(\chi_{\mathbb{1}}(\bar{\tau}) - \chi_\psi(\bar{\tau})) + \text{c.c.}.$$

(123)

Here the partition $Z_{\sigma|\sigma}$ twisted by $D_\sigma$ in both space and time directions is defined with the resolved topological junction via $D_x = D_{\mathbb{1}}$ in figure 18. One can check that the above is consistent with the modular transformations (122) using the modular $S$-matrix for $\text{Vir}_{c=1/2}$,

$$S_{ij} = \frac{1}{2}\begin{pmatrix} 1 & 1 & \sqrt{2} \\ 1 & 1 & -\sqrt{2} \\ \sqrt{2} & -\sqrt{2} & 0 \end{pmatrix}.$$

(124)

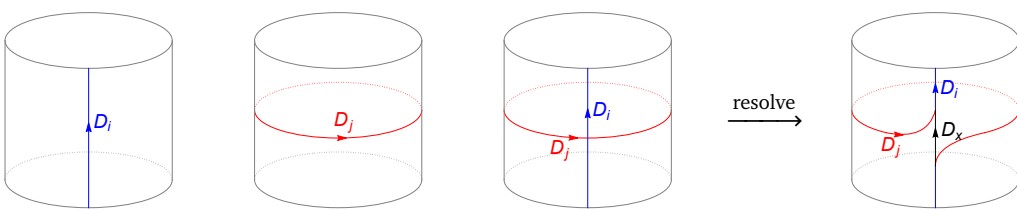

Figure 18: Torus partition functions twisted by various configurations of topological defects. The third diagram involves a pair of topological defects that intersect at a topological junction. Its resolution in the diagram to the right defines the twisted partition $Z_{i|j}$ in the text (the dependences on the resolution channel labelled by defect $D_x$ and the pair of three-fold junctions are implicit). See also Fig. 16 for the action of topological defects on point-like operators.

## C.2  Ising topological defects from the Majorana fermion

The Ising CFT is the bosonization of the Majorana fermion. Correspondingly, we expect generalized symmetries (described by the Ising fusion category) in the Ising CFT to have origins in the fermionic theory.

We start by reviewing the symmetries of a free Majorana fermion $(\psi, \bar{\psi})$. There is a non-chiral $\mathbb{Z}_2^F$ fermion number symmetry generated by $(-1)^F$ and a chiral parity symmetry $\mathbb{Z}_2^L$

$$(-1)^{F_L}: \quad \psi(z) \to -\psi(z), \quad \bar{\psi}(\bar{z}) \to \bar{\psi}(\bar{z}).$$

(125)

As we will explain, under the bosonization map, the "dual symmetry" of $(-1)^F$ becomes the $\mathbb{Z}_2^C$ symmetry generator $D_\psi$ and $(-1)^{F_L}$ becomes the Kramers-Wannier duality defect $D_\sigma$.

In the fermionic theory, depending on the spin structure (fermion boundary condition) on the spatial $S^1$, we have two Hilbert spaces $\mathcal{H}_{\text{NS}}$ (for periodic boundary condition) and $\mathcal{H}_{\text{R}}$ (for anti-periodic boundary condition). They further decompose into even and odd subsectors with respect to $(-1)^F$, and we list the Virasoro primaries in each sector below,

$$\mathcal{H}_{\text{NS}}^+ = \{\mathbb{1}, \epsilon = i\psi\bar{\psi}\}, \ \mathcal{H}_{\text{NS}}^- = \{\psi, \bar{\psi}\}, \ \mathcal{H}_{\text{R}}^+ = \{\sigma\}, \ \mathcal{H}_{\text{R}}^- = \{\mu\}. \tag{126}$$

The Ising CFT is obtained from the Majorana fermion by bosonization (the continuum version of the Jordan-Wigner transformation). The relation between the operators in the fermionic theory and the bosonic theory is already implied by our notations above. More explicitly, recall that the bosonization (also known as GSO projection) of a fermionic theory, in the modern point of view, corresponds to gauging a non-anomalous $\mathbb{Z}_2^F$ symmetry [96–98], which involves summing over spin structures (corresponding to $\mathbb{Z}_2^F$ twists) and projecting onto the $\mathbb{Z}_2^F$ invariant subspace. The bosonization procedure is not unique in general. The freedom comes from the choice of the non-anomalous fermion number symmetry and the possibility to stack a fermionic SPT. Here the single Majorana fermion has a unique non-anomalous $\mathbb{Z}_2^F$ symmetry and the only freedom in bosonization comes from stacking a fermionic SPT defined by the Arf invariant [99].[4]

At the level of the partition function on a genus $g$ Riemann surface, the Ising CFT and the Majorana fermion are related by [96–98]

$$Z_{\text{Ising}}(A) = \frac{1}{2^g} \sum_\rho Z_{\text{Majorana}}(\rho)(-1)^{\text{Arf}(\rho+A)}, \tag{127}$$

where $\rho$ denotes the spin structure and $A$ is the background $\mathbb{Z}_2^C$ gauge field. The sum over $\rho$ on the RHS gauges the $(-1)^F$ symmetry of the Majorana fermion, and $\mathbb{Z}_2^C$ emerges as the corresponding "quantum symmetry" or "dual symmetry" in the Ising CFT. This is a common feature in $1+1d$ for orbifolds by a discrete abelian group $G$ and the "dual symmetry" charges are $G$-Wilson lines that measure the magnetic flux in the $G$-twisted sectors [100]. Note that the other possible bosonization map from stacking the fermionic SPT $(-1)^{\text{Arf}(\rho)}$

$$Z_{\text{Ising}'}(A) = \frac{1}{2^g} \sum_\rho Z_{\text{Majorana}}(\rho)(-1)^{\text{Arf}(\rho+A)+\text{Arf}(\rho)}. \tag{128}$$

produces the same (isomorphic) Ising CFT (in particular $Z_{\text{Ising}'}(A) = Z_{\text{Ising}}(A)$) because the Majorana satisfies [99]

$$Z_{\text{Majorana}}(\rho) = Z_{\text{Majorana}}(\rho)(-1)^{\text{Arf}(\rho)}. \tag{129}$$

This property is a consequence of the anomalous $(-1)^{F_L}$ symmetry of the Majorana fermion and is closely related to the duality symmetry in the Ising CFT which we will come to next.

To see more explicitly how the chiral $(-1)^{F_L}$ symmetry of the Majorana fermion descends to the duality defect $D_\sigma$ in the Ising CFT, we first note that the energy operator $\epsilon = i\psi\bar{\psi}$ is odd under $(-1)^{F_L}$, which matches with how $D_\sigma$ acts on $\epsilon$ from Table 1. Furthermore $(-1)^{F_L}$ of the Majorana fermion carries a $\mathbb{Z}_8$ anomaly [101–104]. This is a fermionic mixed gravitational anomaly classified by the cobordism group $\Omega_{\text{spin}}^3(B\mathbb{Z}_2^{F_L}) = \mathbb{Z}_8$ and the single Majorana fermion provides a generator for this $\mathbb{Z}_8$ anomaly. The anomaly can be detected in several ways, for example by studying the Hilbert space $\mathcal{H}_{\text{R}}$ on $S^1$ with periodic boundary condition, where the $\mathbb{Z}_2^{F_L}$ and $\mathbb{Z}_2^F$ generators anti-commute (see e.g. [105]),

$$\{(-1)^{F_L}, (-1)^F\} = 0. \tag{130}$$

---

[4]The Arf invariant Arf$(\rho)$ is a $\mathbb{Z}_2$-valued function of the spin structure $\rho$. For example on a torus, Arf$(\rho)$ is non-trivial only when $\rho$ corresponds to the odd spin structure (periodic boundary conditions in both space and time directions).

In particular from (126), this requires that $(-1)^{F_L}$ must exchange $\sigma$ and $\mu$ which have opposite $(-1)^F$ charges, as expected for the Kramers-Wannier duality. The $\mathbb{Z}_8$ anomaly also implies the following relation for the massive Majorana fermion,

$$Z_{\text{Majorana}}(\rho, m) = Z_{\text{Majorana}}(\rho, -m)(-1)^{\text{Arf}(\rho)}, \tag{131}$$

and the massless case $m = 0$ leads to the relation (129) which says that the massless Majorana partition function must vanish whenever $(-1)^{\text{Arf}(\rho)} \neq 1$.

As before, we can record the properties of the fermionic symmetry defects (i.e. their defect Hilbert spaces and the symmetry actions therein) by the twisted partition functions on the torus. We start with the partition functions in the NS and R sectors respectively,

$$Z^F_{\text{NS}}(\tau, \bar{\tau}) \equiv Z^F_{\mathbb{1}|\mathbb{1}}(\tau, \bar{\tau}) = |\chi_{\mathbb{1}}(\tau) + \chi_{\psi}(\tau)|^2, \quad Z^F_{\text{R}}(\tau, \bar{\tau}) \equiv Z^F_{(-1)^F|\mathbb{1}}(\tau, \bar{\tau}) = 2|\chi_{\sigma}(\tau)|^2. \tag{132}$$

In the presence of an additional symmetry defect along the spatial cycle, we have

$$
\begin{aligned}
Z^F_{\mathbb{1}|(-1)^F}(\tau, \bar{\tau}) &= |\chi_{\mathbb{1}}(\tau) - \chi_{\psi}(\tau)|^2, \\
Z^F_{\mathbb{1}|(-1)^{F_L}}(\tau, \bar{\tau}) &= (\chi_{\mathbb{1}}(\tau) - \chi_{\psi}(\tau))(\chi_{\mathbb{1}}(\bar{\tau}) + \chi_{\psi}(\bar{\tau})), \\
Z^F_{\mathbb{1}|(-1)^{F_R}}(\tau, \bar{\tau}) &= (\chi_{\mathbb{1}}(\tau) + \chi_{\psi}(\tau))(\chi_{\mathbb{1}}(\bar{\tau}) - \chi_{\psi}(\bar{\tau})), \\
Z^F_{(-1)^{F_L}|\mathbb{1}}(\tau, \bar{\tau}) &= \sqrt{2}\chi_{\sigma}(\tau)(\chi_{\mathbb{1}}(\bar{\tau}) + \chi_{\psi}(\bar{\tau})), \\
Z^F_{(-1)^{F_L}|(-1)^{F_L}}(\tau, \bar{\tau}) &= \sqrt{2}e^{\frac{\pi i}{8}}\chi_{\sigma}(\tau)(\chi_{\mathbb{1}}(\bar{\tau}) - \chi_{\psi}(\bar{\tau})), \\
Z^F_{(-1)^{F_R}|\mathbb{1}}(\tau, \bar{\tau}) &= \sqrt{2}(\chi_{\mathbb{1}}(\tau) + \chi_{\psi}(\tau))\chi_{\sigma}(\bar{\tau}), \\
Z^F_{(-1)^{F_R}|(-1)^{F_R}}(\tau, \bar{\tau}) &= \sqrt{2}e^{-\frac{\pi i}{8}}(\chi_{\mathbb{1}}(\tau) - \chi_{\psi}(\tau))\chi_{\sigma}(\bar{\tau}), \\
Z^F_{(-1)^F|(-1)^F}(\tau, \bar{\tau}) &= Z^F_{(-1)^F|(-1)^{F_L}}(\tau, \bar{\tau}) = Z^F_{(-1)^F|(-1)^{F_R}}(\tau, \bar{\tau}) = 0, \\
Z^F_{(-1)^{F_L}|(-1)^F}(\tau, \bar{\tau}) &= Z^F_{(-1)^{F_R}|(-1)^F}(\tau, \bar{\tau}) = Z^F_{(-1)^{F_R}|(-1)^{F_L}}(\tau, \bar{\tau}) = 0.
\end{aligned}
\tag{133}
$$

Here $(-1)^{F_R}$ is the right-moving fermion parity. The above partition functions are related to the twisted partition functions (123) in the Ising CFT by the bosonization map (127). The vanishing of the twisted partition functions in the last two lines of (133) is again a consequence of the $\mathbb{Z}_8$ anomaly (129) (see also below (131)). From the above, we also observe that the precise relation between the duality defect in the Ising CFT, and the $(-1)^{F_L}$ symmetry defect in the Majorana fermion is,[5]

$$D_{\sigma} \leftrightarrow \sqrt{2}(-1)^{F_L}. \tag{134}$$

## C.3   Majorana fermion in the duality-twisted sector

In the main text, we have identified a Majorana zero mode in the duality-twisted Floquet circuit and showed that this Majorana mode is a symmetry of the circuit. Here we discuss its counterpart in the continuum both at the critical point and away from criticality. We will see that the Majorana zero mode originates from the Majorana fields $\psi(x)$ (and $\bar{\psi}(x)$) attached to the spin-flip defect $D_{\psi}$ that anchors topologically on the duality defect $D_{\sigma}$ (see figure 19).

We start at the critical point. First of all, from the F-moves (as in figure 15) using the F-symbols in (118), it is easy to see that the Majorana field $\psi(x), \bar{\psi}(x)$ commutes the spin-flip defect $D_{\psi}$ in the duality twisted sector (see figure 20). Next let us see how quantization of the Majorana fields leads to the spectrum in the defect-twisted Hilbert space $\mathcal{H}_{\sigma}$ in (119). We work with the cylinder spacetime with coordinates $(x_1, x_2) \sim (x_1, x_2 + 2\pi)$. States in $\mathcal{H}_{\sigma}$

---

[5]More precisely this relation involves a topological interface $\mathcal{I}$ between the bosonic Ising CFT on the left and the Majorana fermion on the right such that $D_{\sigma}\mathcal{I} = \mathcal{I}\sqrt{2}(-1)^{F_L}$ [106].

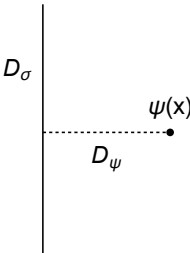

Figure 19: The Majorana field $\psi(x)$ (similarly for $\bar{\psi}(x)$) attached to the spin flip defect $D_\psi$ that joins the duality defect $D_\sigma$ at a topological junction.

has eigenvalue $\pm i$ under the spin-flip symmetry which acts as $\hat{D}_\psi$ (see figure 16 and Table 1), correspondingly we have the obvious decomposition $\mathcal{H}_\sigma = \mathcal{H}_\sigma^+ \oplus \mathcal{H}_\sigma^-$. As a consequence of the F-moves as in figure 21, we find that the Majorana fields have the following periodicity conditions in the two sectors,

$$
\begin{aligned}
\mathcal{H}_\sigma^+ &: \ \psi(x_1, x_2 + 2\pi) = \psi(x_1, x_2), \ \bar{\psi}(x_1, x_2 + 2\pi) = -\bar{\psi}(x_1, x_2), \\
\mathcal{H}_\sigma^- &: \ \psi(x_1, x_2 + 2\pi) = -\psi(x_1, x_2), \ \bar{\psi}(x_1, x_2 + 2\pi) = \bar{\psi}(x_1, x_2).
\end{aligned}
\tag{135}
$$

Quantization proceeds by decomposing $\psi, \bar{\psi}$ into Fourier modes respecting the periodicity conditions. Each sector has a ground state whose energy and momentum are determined by standard Casimir energy analysis and are here given by [107],

$$
(E, P) = \left( \frac{1}{48}, \frac{1}{16} \right), \quad \left( \frac{1}{48}, -\frac{1}{16} \right).
\tag{136}
$$

Using the relation between cylinder and plane conformal charges,

$$
E = L_0 + \bar{L}_0 - \frac{c}{12}, \qquad P = L_0 - \bar{L}_0,
\tag{137}
$$

we see the two states in (136) correspond to precisely the two operators $s, \bar{s}$ listed in (119) respectively. We note that the periodicity conditions (135) allows for a zero mode for $\psi$ in the $\mathcal{H}_\sigma^+$ sector and a zero mode for $\bar{\psi}$ in the $\mathcal{H}_\sigma^-$ sector. They preserve the ground states in their respective sector (up to a sign that can be reabsorbed into the definition of $\psi, \bar{\psi}$). The states of higher energy in $\mathcal{H}_\sigma$ are obtained from these ground states by creation operators from the Fourier expansion of $\psi, \bar{\psi}$.

The Ising CFT does not admit relevant translation invariant perturbations that commute with the duality defect. Nonetheless, we can turn on a spatially inhomogeneous coupling to the energy operator

$$
S_{\text{CFT}} \rightarrow S_{\text{CFT}} + \int d^2 x \, m(x) \epsilon(x),
\tag{138}
$$

while preserving the topological defects. This is achieved by choosing a domain-wall profile for $m(x)$ centered at $x_2 = 0$ with width $2\ell$ such that (see figure 22)

$$
m(x) = \begin{cases} M & x_2 \in (\ell, w), \\ -M & x_2 \in (w, 2\pi - \ell), \end{cases}
\tag{139}
$$

where we insert the duality defect $D_\sigma$ at $x_2 = w$, which is by construction topological as long as $\ell < w < 2\pi - \ell$ (see figure 22). The Majorana fields now become massive and contribute a single localized Majorana zero mode at the domain wall (for $M > 0$ from $\psi(x)$ as in figure 22 and for $M < 0$ from $\bar{\psi}(x)$) [108]. The setup here corresponds to the small coupling and zero

temperature limit of the Floquet circuit considered in Section 5.1, and we find the continuum description of the Majorana zero mode identified there.

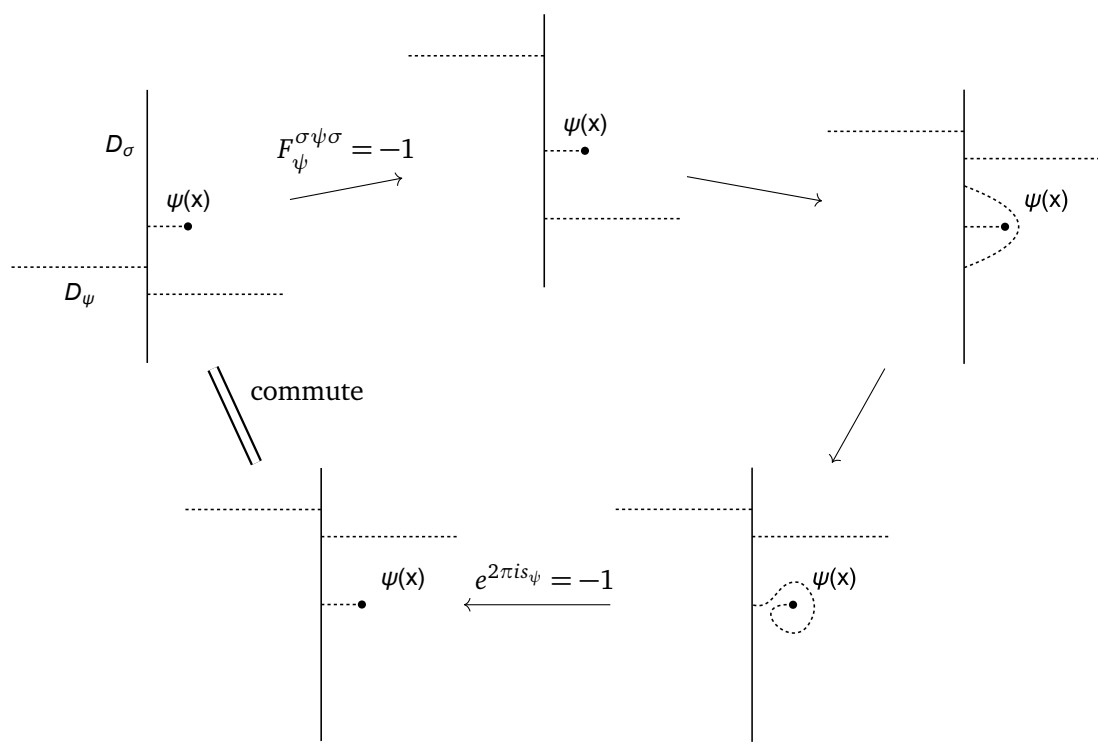

Figure 20: A sequence of F-moves (clockwise from top-left to bottom-left) that shows $D_\psi$ commutes with the Majorana field $\psi(x)$ (similarly for $\bar\psi(x)$) in the presence of the duality twist by $D_\sigma$. The only step where the F-move introduces a nontrivial factor is the first step on the top-left. The $-1$ contribution there is cancelled by the $-1$ from $2\pi$-rotation of the spin-$\frac{1}{2}$ $\psi(x)$ field in the last step at the bottom.

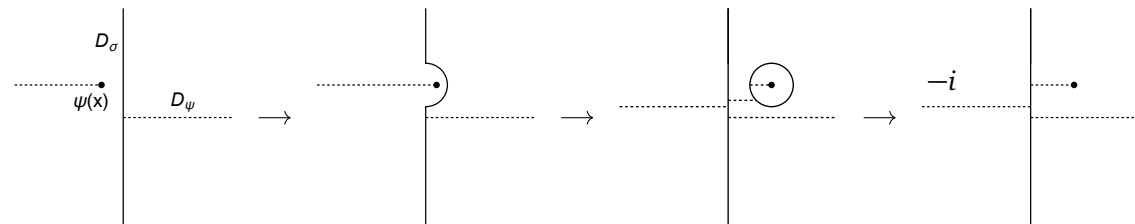

Figure 21: Periodicity of the Majorana field $\psi(x)$ in the duality twisted sector from a sequence of F-moves. The left and right ends of each diagram are identified (cylinder topology). There is a nontrivial factor $(F_\sigma^{\sigma\sigma})_{\mathbb{1}\psi} = \frac{1}{\sqrt{2}}$ from the second step and then a $-i\sqrt{2}$ factor from $\hat{D}_\sigma\psi(x)$ in the third step. Together they imply that bringing $\psi(x)$ around the cylinder generates a horizontal $D_\psi$ defect and introduces a factor of $-i$. Similarly, performing these moves for $\bar\psi(x)$ gives the same right-most diagram but with a factor of $i$. The duality twisted sector decomposes $\mathcal{H}_\sigma = \mathcal{H}_\sigma^+ \oplus \mathcal{H}_\sigma^-$ according to charges for $D_\psi$ which can be $\pm i$ (see above (135)). Together with the $-i$ factor from the F-move, this determines the periodicity for $\psi(x)$ (and similarly for $\bar\psi(x)$).

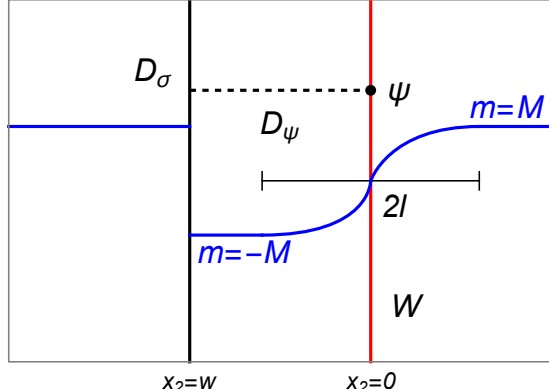

Figure 22: Ising CFT on the cylinder twisted by the duality defect $D_\sigma$ in the presence of a domain wall profile (blue) for the mass coupling $m(x)$ in (138). Here the domain wall $W$ (red) has width $2\ell$ and $M > 0$. The domain wall Majorana mode comes from the $\psi(x)$ field attached to the spin flip defect $D_\psi$.

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
