# Peer review of "Topological Defects in Floquet Circuits"

_SciPost Physics, doi:SciPost Phys. 16, 075 (2024)_

## Round 1 · Referee Report · Anonymous (Referee 1) · 2023-8-4

Strengths
1. Introducing novel quantum field theory notions into non-equilibrium systems.
2. Paper is well written. Through explicit construction, the idea of categorical symmetry is introduced in a pedagogical way.
Weaknesses
The motivation and physical implementation needs to be further clarified.
Report
In this paper, the authors introduced novel quantum field theory notions like categorical symmetry and duality defect to the design of Floquet circuits. In previous studies of Floquet systems, circuits based on the transverse field Ising model played a major role in the discovery of for example Floquet time crystal. In the equilibrium transverse field Ising model, it is well known that the model is self-dual and the duality transformation has been studied recently as an example of categorical symmetry in the system. By introducing such symmetry and related defect into the design of the Floquet circuit, the authors were able to observe the same `topological' nature of the symmetry as in equilibrium systems, reproduce the same categorical fusion structure, and introduce Majorana zero modes into Floquet systems. There are a few questions that I hope the authors can address.
1. The duality transformation is not unitary. Maybe I missed it, but it is not entirely clear how to implement the transformation or to insert the twisted boundary in the Floquet circuit. Is measurement a necessary part of the protocol? Does one need to post-select based on the measurement result?
2. While introducing topological defects into Floquet systems induces interesting features, most of the added features seem to reproduce what one expects in the equilibrium version of the system. Is this the general expectation? Could topological defect in Floquet system have new properties than their equilibrium counterpart?
3. It is not entirely clear whether the features describes in this paper are stable against errors in the implementation of the circuit.
If these questions are adequately addressed, I think this paper is worth publishing in SciPost Physics as it connects two very different fields and introduces some new ideas.

---

## Round 1 · Referee Report · Anonymous (Referee 2) · 2023-8-7

Strengths
1. The writing is clear.
2. The results presented are original and innovative.
3. The study of topological defects in Floquet circuits presented in this work could offer essential information for the understanding and classification of Floquet symmetry-protected topological phases.
Weaknesses
Sec. V reads a bit too technical.
Report
In this work, the authors investigated Floquet circuits with topological defects, which obey non-trivial fusion rules and can be deformed in space and time without changing the physics. Circuits with topological defects for the Floquet Ising model are constructed under different boundary conditions. Specially, the duality-twisted boundary condition allows the system to a host an isolated Majorana zero mode, which was found to manifest in the auto-correlation function.
Overall, this manuscript is clearly written and the results presented are innovative. This detailed study of topological defects in Floquet circuits could offer indispensable information for the understanding and classification of Floquet symmetry-protected topological phases. I think this manuscript could be accepted by Scipost Physics for publication after the following questions are addressed.
Requested changes
1. A section or subsection could be added to discuss how to realize the proposed Floquet circuits in experiments and how to detect the related topological properties. Specially, are there any suggestions for the realization of the duality-twisted boundary conditions and the isolated Majorana zero modes in any real experimental setups?
2. One unique feature of the Floquet Ising-Majorana chain is to hold Majorana edge modes with the quasienergy pi. Is there any possibility to find a single unpaired localized Majorana pi mode in the Floquet circuits considered by the authors? Under which conditions could we obtain Floquet pi modes in Floquet circuits with topological defects? The authors are suggested to comment on these issues.

---

## Round 2 · Referee Report · Anonymous (Referee 2) · 2024-2-26

Report

The authors have addressed my concerns and improved their manuscript accordingly. I think the present version of the manuscript is suitable for publication in Scipost Physics.

---

## Round 2 · Author Response

Dear Referees and Editor,

Thank you for considering our manuscript and for the many positive
comments. Below, we respond to all the questions raised by the Referees, and
list the changes made to the manuscript.

Response to Referee 1

Q1: The duality transformation is not unitary. Maybe I missed it, but it is not entirely clear how to implement the transformation or to insert the twisted boundary in the Floquet circuit. Is measurement a necessary part of the protocol? Does one need to post-select based on the measurement result?

Our Response:
Inserting a duality defect in the space-like direction is not
straightforward, and there are
several suggestions that involve implementing the duality defect by unitaries and measurements \cite{tantivasadakarn2021long}. In the first part of the paper, where we simulate the duality defect in the space-like direction, and demonstrate fusion rules, we implement the duality defect by following the rules of category theory, which in particular produce a restricted set of linear transformations on the Hilbert space, not-necessarily unitary. Explicitly, the duality defect can be defined by its action on a basis as seen in (10) and produces states living on the dual lattice and does not in principle require measurements although measurements could be used to implement it as explained in \cite{tantivasadakarn2021long}. In their scheme, post-selection is not required to implement the Kramers-Wannier transformation as they can simply apply Pauli X matrices on the dual lattice depending on the measurement outcome to ensure that the Kramers-Wannier duality transformation is implemented correctly.

For the second part of the paper, when we consider the duality twist, the rules of category theory give us a modified time-evolution operator, which is perfectly unitary and can be simulated by two-site unitaries (as was done in a follow-up study on the quantum computer
(see response to Referee 2 below).

There is no local unitary transformation that can transform the untwisted unitary to the twisted unitary, and whether they can be transformed into one another through measurements, is an interesting open question.

Q2:While introducing topological defects into Floquet systems induces interesting features, most of the added features seem to reproduce what one expects in the equilibrium version of the system. Is this the general expectation? Could topological defect in Floquet system have new properties than their equilibrium counterpart?

Our Response:
This is an interesting open question that we plan to explore in the future.

Q3: It is not entirely clear whether the features describes in this paper are stable against errors in the implementation of the circuit.

Our Response:
If there are errors, then indeed, the exact commutation relations of the defects will not be obeyed, and in the long time and large system size limit, the localized zero mode will become unstable. However, if the system size is small, then the localized Majorana can still be stable even at infinite times. This was shown in a follow up study where perturbing terms were included (https://doi.org/10.1103/PhysRevB.107.245416). Two kinds of perturbing terms were considered, those that commute with the $Z_2$ symmetry, and those that did not. It was then shown that for the zero mode to decay, the chain has to act like an effective reservoir, which requires going to large system sizes.

Response to Referee 2

Q1: A section or subsection could be added to discuss how to realize the proposed Floquet circuits in experiments and how to detect the related topological properties. Specially, are there any suggestions for the realization of the duality-twisted boundary conditions and the isolated Majorana zero modes in any real experimental setups?

Our Response:
The defects can be realized in current noisy intermediate scale quantum devices. The duality twisted unitary, for example, was recently implemented in the following follow-up study (https://doi.org/10.48550/arXiv.2308.02387).

Q2: One unique feature of the Floquet Ising-Majorana chain is to hold Majorana edge modes with the quasienergy pi. Is there any possibility to find a single unpaired localized Majorana pi mode in the Floquet circuits considered by the authors? Under which conditions could we obtain Floquet pi modes in Floquet circuits with topological defects? The authors are suggested to comment on these issues.

Our Response:
We note that it is not possible to realize an isolated $\pi$ mode in the setup presented in the manuscript. This is explained on page 25-26 of the manuscript. The reason is that the determinant of the unitary here has to be unity. Since the unitary is odd-dimensional, this rules out an isolated eigenvalue of $-1$ which would correspond to an isolated $\pi$ mode.

---

## Round 2 · List of Changes

1. Typos corrected.
  2. References updated and added.
  3. A silly error, where site indices where mis-labelled in Eq. 23(b), has been corrected.
  4. On page 8, and at the end of the second paragraph, we point out that a brief discussion of $\pi$ modes will be presented later, so that the reader anticipates it.
  5. In Section 3.2, in response to the first question of Referee-1, we give some more details on how the system was numerically simulated.
  6. A discussion has been added in the second paragraph on Page 16, on the non-local nature of the transformation between the defectless unitary and the duality twisted unitary.
  7. In the conclusions, we have added discussion on both experimental realization as well as robustness to perturbations.

---

## Editorial Decision

published